# Epigenetic interaction between UTX and DNMT1 regulates diet-induced myogenic remodeling in brown fat

Fenfen Li[1], Jia Jing[1], Miranda Movahed[1], Xin Cui [1], Qiang Cao[1], Rui Wu[1], Ziyue Chen[2], Liqing Yu[3], Yi Pan[2,4], Huidong Shi [5,6], Hang Shi[1✉] & Bingzhong Xue [1✉]

Brown adipocytes share the same developmental origin with skeletal muscle. Here we find that a brown adipocyte-to-myocyte remodeling also exists in mature brown adipocytes, and is induced by prolonged high fat diet (HFD) feeding, leading to brown fat dysfunction. This process is regulated by the interaction of epigenetic pathways involving histone and DNA methylation. In mature brown adipocytes, the histone demethylase UTX maintains persistent demethylation of the repressive mark H3K27me3 at *Prdm16* promoter, leading to high *Prdm16* expression. PRDM16 then recruits DNA methyltransferase DNMT1 to *Myod1* promoter, causing *Myod1* promoter hypermethylation and suppressing its expression. The interaction between PRDM16 and DNMT1 coordinately serves to maintain brown adipocyte identity while repressing myogenic remodeling in mature brown adipocytes, thus promoting their active brown adipocyte thermogenic function. Suppressing this interaction by HFD feeding induces brown adipocyte-to-myocyte remodeling, which limits brown adipocyte thermogenic capacity and compromises diet-induced thermogenesis, leading to the development of obesity.

[1] Department of Biology, Georgia State University, Atlanta, GA 30303, USA. [2] Department of Computer Science, Georgia State University, Atlanta, GA 30303, USA. [3] Division of Endocrinology, Diabetes and Nutrition, Department of Medicine, University of Maryland School of Medicine, Baltimore, MD 21201, USA. [4] Shenzhen Institute of Advanced Technology, Chinese Academy of Sciences, Shenzhen 518055, P.R. China. [5] Georgia Cancer Center, Medical College of Georgia, Augusta University, Augusta, GA 30912, USA. [6] Department of Biochemistry and Molecular Biology, Medical College of Georgia, Augusta University, Augusta, GA 30912, USA. ✉email: hshi3@gsu.edu; bxue@gsu.edu

Obesity is now considered as an epidemic disorder that poses as an independent risk factor for the development of various metabolic disorders such as insulin resistance/type 2 diabetes, hypertension, dyslipidemia, and cardiovascular diseases[1]. Chronic energy excess due to energy intake over energy expenditure results in obesity[1]. Thus, understanding the mechanism underlying the regulation of energy homeostasis may help identify the therapeutic targets for the prevention and treatment of obesity.

Total energy expenditure consists of three aspects: energy required for basal metabolic rate, energy expended to perform physical activity and energy used to generate heat. The last one is termed as adaptive thermogenesis, which mainly takes place in brown adipose tissue (BAT) due to the unique presence of uncoupling protein 1 (UCP1) in the inner membrane of mitochondria[1,2]. UCP1 acts to uncouple oxidative phosphorylation from ATP synthesis, thereby dissipating energy as heat and profoundly increasing overall energy expenditure[3,4]. In addition, recent studies have reported the existence of UCP1-independent thermogenesis that is generated by sarco/endoplasmic reticulum $Ca^{2+}$-ATPase 2b/ATPase, $Ca^{2+}$ transporting, cardiac muscle, slow twitch 2 (SERCA2b/ATP2a2)-mediated calcium cycling or creatine-driven substrate cycling[5,6]. Given the presence of human brown fat that profoundly increases energy expenditure[7–9], better understanding the mechanism underlying BAT thermogenesis has a translational implication for the treatment of obesity.

Most of the current studies investigating the mechanisms in the regulation of brown fat thermogenesis focus on cellular signaling pathways; much less is known about the epigenetic mechanisms in this process. We have recently discovered several epigenetic pathways involved in adipocyte development and brown adipocyte thermogenesis[10–13]. For instance, we have reported that ubiquitously transcribed tetratricopeptide/lysine (K)-specific demethylase 6A (Utx/Kdm6a), a histone demethylase that preferentially catalyzes the demethylation of tri-methylated histone H3 lysine 27 (H3K27me3) and therefore relieves its ability of gene silencing[14], plays a key role in regulating brown adipocyte thermogenic program via a coordinated regulation of H3K27 demethylation and acetylation[13]. Specifically, Utx, whose expression in white adipose tissue (WAT) or BAT is induced by cold exposure, acts as a positive regulator of BAT thermogenic gene expression[13]. However, the physiological significance of Utx in the regulation of energy homeostasis remains unknown. In this study, we have generated mice with brown adipocyte-specific Utx knockout (UTXKO) and characterized metabolic phenotypes of these mice. We have further interrogated potential epigenetic mechanisms underlying Utx's regulation of brown fat function during diet-induced obesity (DIO), and found that this process involves an interaction between histone and DNA methylation in the promoters of key molecules regulating brown or myogenic lineage determination, leading to a myogenic remodeling and thermogenic dysfunction in BAT of UTXKO mice during the development of DIO. We have also identified that BAT-to-myocyte remodeling in BAT represents a common feature in DIO, which eventually leads to BAT dysfunction and contributes to the development of DIO.

## Results

**Mice with UTX deficiency in brown fat exhibit impaired BAT thermogenesis and are susceptible to DIO.** We previously reported that Utx knockdown reduces mRNA levels of brown-specific genes, whereas overexpression of Utx does the opposite in cultured brown adipocyte cell lines[13]. However, it remains unknown whether Utx regulates BAT thermogenic function and whole-body energy homeostasis in vivo. In the current study, we first measured Utx expression pattern in brown and white adipose tissues. Utx mRNA level was higher in interscapular brown adipose tissue (iBAT) than in inguinal white adipose tissue (iWAT) and epididymal WAT (eWAT), and was induced in both BAT and WAT by a 7-day 5 °C cold challenge (Supplementary Fig. 1A and 1B).

To determine the role of Utx in the regulation of BAT thermogenic function and energy homeostasis in vivo, we generated mice with specific deletion of Utx in brown fat (UTXKO) by crossing Utx-floxed mice[15] with Ucp1-cre mice[16]. Utx is located on the X chromosome but escapes X chromosome inactivation in females[17]. Thus, female UTXKO mice were defined as homozygous $Utx^{fl/fl}$ with Ucp1-Cre ($Ucp1-Cre::Utx^{fl/fl}$), with $Utx^{fl/fl}$ littermates as control; whereas male UTXKO mice were defined as hemizygous $Utx^{fl/Y}$ with Ucp1-Cre ($Ucp1-Cre::Utx^{fl/Y}$), with $Utx^{fl/Y}$ littermates as control. Male and female UTXKO and their fl/Y or fl/fl littermate control mice were viable and were born with expected Mendelian frequency, and as expected, Utx deletion in brown fat resulted in an 80% reduction of Utx mRNA expression in male UTXKO mice compared to their fl/Y littermates (Supplementary Fig. 2A).

Although there was no difference in body weight of male UTXKO mice compared to that of fl/Y mice fed a regular chow diet (Supplementary Fig. 2B), UTXKO mice exhibited a significant increase in fat mass and a significant decrease in lean mass (Supplementary Fig. 2C) measured by a Minispec NMR body composition analyzer. In consistence, we also found significantly increased iWAT, and a tendency of increased eWAT mass in male UTXKO mice (Supplementary Fig. 2D). This was associated with larger adipocytes in iBAT and iWAT (Supplementary Fig. 2E) and less UCP1 immunohistochemistry (IHC) staining in iBAT (Supplementary Fig. 2F), suggesting a potential decrease in brown fat thermogenesis in brown fat of UTXKO mice compared to their fl/Y littermates. With increased adiposity, male UTXKO mice exhibited an increase in fed and fast glucose levels and fed insulin levels (Supplementary Fig. 2G), suggesting that these UTXKO mice had impaired insulin sensitivity compared to their fl/Y littermate controls. Indeed, this was further confirmed by impaired glucose tolerance test (GTT) (Supplementary Fig. 2H) with increased circulating insulin levels at 15 min during GTT in male UTXKO mice compared to that of fl/Y mice (Supplementary Fig. 2I).

To determine the role of brown fat UTX in the regulation of diet-induced obesity, we conducted metabolic characterization of body weight, energy metabolism and insulin sensitivity in UTXKO mice fed a high-fat diet (HFD). As shown in Fig. 1A, male UTXKO mice gained significantly more weight compared to their fl/Y littermates on the HFD. Using a Minispec NMR body composition analyzer, we found that UTXKO mice exhibited a significant increase in fat mass and a significant decrease in lean mass (Fig. 1B). In consistence, iWAT and iBAT mass was also significantly increased in UTXKO mice compared to their fl/Y littermates (Fig. 1C) with larger adipocytes in iBAT, iWAT, and eWAT (Fig. 1D). Using a PhenoMaster metabolic cage system, we found that UTXKO mice displayed lower energy expenditure and oxygen consumption (Fig. 1E and Supplementary Fig. 3A), whereas there was no difference in locomotor activity, respiratory exchange rate (RER), and food intake between UTXKO and fl/Y mice (Supplementary Fig. 3B–D). These data indicate that reduced energy expenditure may primarily account for the obese phenotype in UTXKO mice fed the HFD. Quantitative RT-PCR analysis revealed significantly reduced thermogenic gene expression, including Ucp1, PR domain containing 16 (Prdm16), peroxisome proliferative activated receptor gamma coactivator 1 alpha (Pgc1α), Pgc1β, early B cell factor 2 (Ebf2), Ebf3, carnitine palmitoyltransferase 1b, muscle (Cpt1b), cytochrome c oxidase

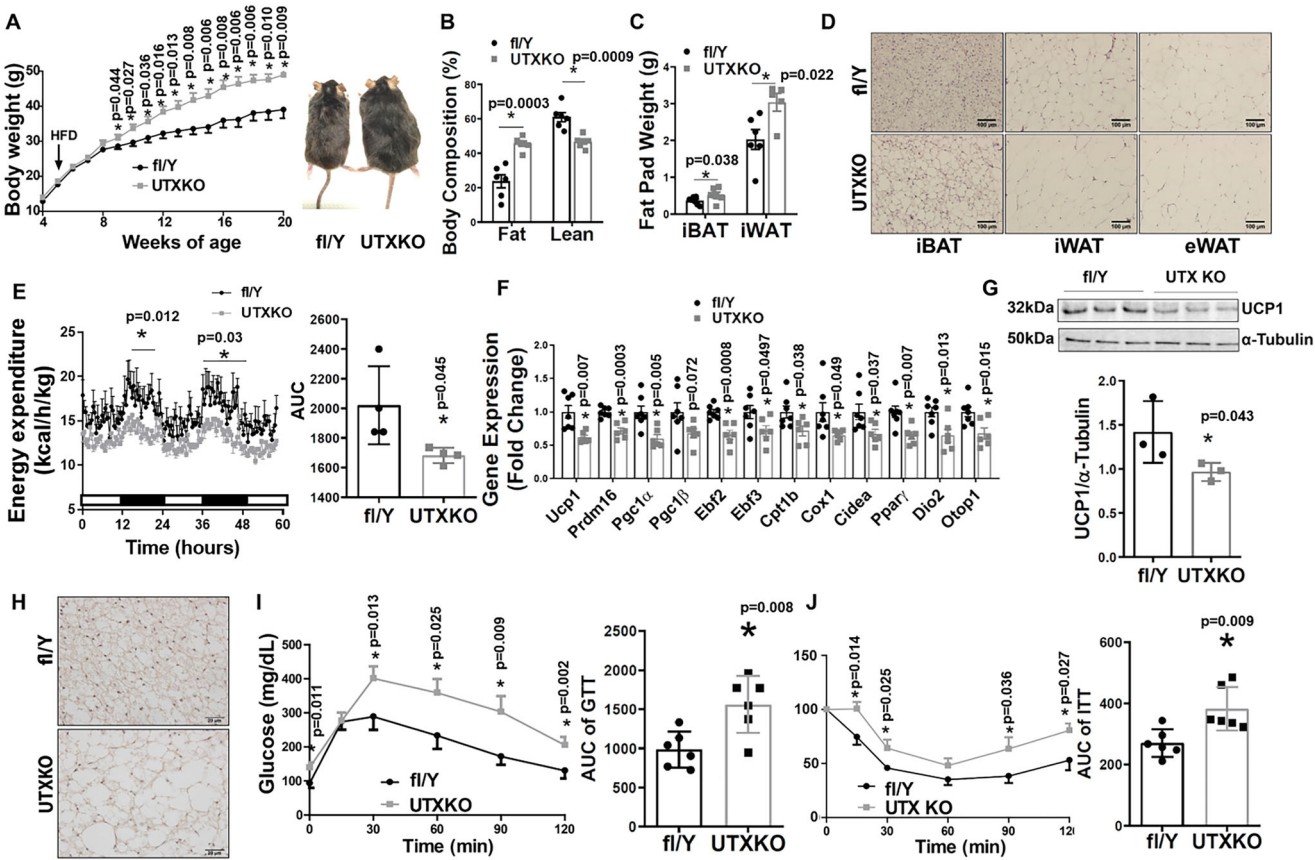

**Fig. 1 UTX deficiency in brown fat promotes high-fat diet (HFD)-induced obesity.** Male UTXKO and their littermate control fl/Y mice were put on HFD when they were 5 weeks of age. **A–C** Body weight growth curve (**A**, fl/Y = 7, UTXKO = 6), Body composition (**B**, n = 6/group), and Fat pad weight (interscapular brown adipose tissue (iBAT) (n = 6/group) and inguinal white adipose tissue (iWAT)(fl/Y = 6, UTXKO = 5) (**C**) in male UTXKO and fl/Y mice fed HFD. *Indicates statistical significance between UTXKO and fl/Y by two-tailed unpaired Student's t-test. **D** Representative H&E staining of iBAT, iWAT, and eWAT in male UTXKO and fl/Y mice fed HFD (n = 3 replicates for each group). **E** Energy expenditure in male UTXKO and fl/Y mice fed HFD (n = 4/group). Left, *indicates statistical significance between fl/fl and D1KO analyzed by ANOVA with repeated measures. Time 17–22 h, F(1,6) = 12.861, p = 0.012; time 36–48 h, F(1,6) = 7.988, p = 0.03. Right, *indicates statistical significance between UTXKO and fl/Y by two-tailed unpaired Student's t-test. **F** Thermogenic gene expression in iBAT measured by quantitative RT-PCR in male UTXKO and fl/Y mice fed HFD (n = 7 for fl/Y and 6 for UTXKO). *indicates statistical significance between UTXKO and fl/Y by two-tailed unpaired Student's t-test. **G** Immunoblotting of UCP1 in iBAT of male UTXKO and fl/Y mice fed HFD (n = 3/group). *indicates statistical significance between UTXKO and fl/Y by two-tailed unpaired Student's t-test. **H** Representative immunohistochemistry (IHC) staining of UCP1 in iBAT of male UTXKO and fl/Y mice fed HFD (n = 3 replicates for each group). **I–J** Glucose tolerance test (GTT) (**I**) (n = 6/group) and Insulin tolerance test (ITT) (**J**) (n = 6/group) in male UTXKO and fl/Y mice fed HFD. *Indicates statistical significance between UTXKO and fl/Y by two-tailed unpaired Student's t-test. All data are expressed as mean ± SEM.

subunit I (*Cox1*), cell death-inducing DNA fragmentation factor, alpha subunit-like effector A (*Cidea*), *Pparγ*, type II deiodinase (*Dio2*) and otopetrin 1 (*Otop1*) (Fig. 1F) in iBAT of UTXKO mice, which was associated with decreased UCP1 protein levels as measured by both immunoblotting and UCP1 IHC staining (Figs. 1G and 1H). UTXKO mice also displayed glucose intolerance and insulin resistance as assessed by glucose and insulin tolerance tests (GTT and ITT, respectively) (Fig. 1I, J). These data indicate that mice with *Utx* deficiency in brown fat have increased adiposity with impaired insulin sensitivity when fed a regular chow diet, and are susceptible to diet-induced obesity with exacerbated insulin resistance when fed HFD.

Recent data suggest that brown adipocytes secrete various "batokines", including neuregulin 4 (*Nrg4*), peptidase M20 domain containing 1 (*Pm20d1*) and myostatin (*Mstn*), which in turn regulate whole-body insulin sensitivity and metabolic function in other organs[18–20]. To investigate whether deletion of *Utx* in brown adipocytes regulates whole-body energy homeostasis and insulin sensitivity via these secreted batokines, we measured *Nrg4*, *Pm20d1*, and *Mstn* expression in iBAT from

UTXKO and fl/Y littermates under either HFD or cold challenge. As shown in Supplementary Fig 4, the expression of *Nrg4* and *Pm20d1* was slightly increased in iBAT of chow-fed UTXKO mice, but was slightly decreased in iBAT of HFD-fed UTXKO mice compared to that of fl/Y mice (Supplementary Fig. 4A); whereas the expression of *Nrg4* and *Pm20d1* did not change during a 7-day cold challenge between UTXKO and fl/Y littermates (Supplementary Fig. 4B). In addition, whereas *Mstn* expression in iBAT was significantly upregulated by HFD but downregulated by cold exposure, there was no difference in iBAT *Mstn* expression between UTXKO and fl/Y mice (Supplementary Fig 4C–D). Thus, our data suggest that brown adipocyte *Utx* deficiency does not significantly alter the expression of batokines including *Nrg4*, *Pm20d1,* and *Mstn*.

In addition, UCP1-independent thermogenesis that involves SERCA2b/ATP2a2-mediated calcium cycling or creatine kinase (CKM)-mediated creatine cycling may also be important in regulating whole-body energy homeostasis[5,6]. However, *Atp2a2* expression in iBAT was not different between UTXKO and fl/Y littermates under either HFD diet or cold challenge

(Supplementary Fig. 5A–B); whereas the expression of *Ckm* in iBAT was significantly increased in UTXKO compared to that of fl/Y mice fed with either chow or HFD or challenged with cold exposure (Supplementary Fig. 5C–E). Thus, our data suggest that these UCP1-independent thermogenic pathways are unlikely to be involved in UTX-regulated thermogenesis in brown fat.

We also studied metabolic phenotypes of female UTXKO mice and their fl/fl littermate controls. Chow-fed female UTXKO mice did not exhibit any differences in body weight, % of fat and lean mass and fat pad weight (Supplementary Fig. 6A–C). HFD-fed female UTXKO mice had a transient increase in body weight between 11 and 17 weeks of age compared to their fl/fl littermates (Supplementary Fig. 6D). At around 20 weeks of age, although the body weight was not different between the groups, female UTXKO mice still exhibited slightly higher fat mass and lower lean mass, and larger gonadal fat pad mass than that of fl/fl mice (Supplementary Fig. 6E–F). Thus, our data suggest that female UTXKO mice exhibit a mild obesity-prone phenotype when fed HFD.

**Utx deficiency in Myf5-expressing brown adipocyte precursor cells does not regulate BAT thermogenesis and energy homeostasis.** Recent studies suggest that brown adipocytes and skeletal muscle cells share a common developmental lineage and are derived from precursor cells that express myogenic factor 5 (*Myf5*)[21]. To investigate whether *Utx* regulates brown adipocyte function at an early developmental stage, we generated mice with *Utx* deletion in *Myf5*-expressing brown adipocyte/myotube precursor cells (MUTXKO) by crossing *Utx*-floxed mice[15] with *Myf5-Cre* mice[22]. As expected, MUTXKO exhibited around 70% deletion of *Utx* mRNA levels in iBAT (Supplementary Fig. 7A). However, unlike UTXKO mice with *Utx* deficiency in mature brown adipocytes (Fig. 1), male mice with *Utx* deficiency in *Myf5*-expressing brown adipocyte/myotube precursor cells had no differences in body weight, body composition, energy expenditure and locomotor activity compared to their fl/Y littermates when fed HFD (Supplementary Fig. 6B–E). In addition, there was no difference in *Ucp1* and *Pgc1α* expression in iBAT between male MUTXKO and fl/Y mice on HFD (Supplementary Fig. 6F). Thus, our data suggest that the regulation of brown adipocyte function by UTX is dependent on brown adipocyte developmental stage; UTX becomes important in maintaining active brown adipocyte function possibly at a later developmental stage, after the precursor cells are committed to the brown adipocyte lineage.

**Utx deficiency in brown fat induces myogenesis.** To determine the molecular mechanism whereby *Utx* deficiency impaired brown fat thermogenesis and promoted diet-induced obesity, we performed RNA-seq analysis using iBAT from male UTXKO and fl/Y mice fed HFD diet for 12 weeks to unbiasedly examine gene expression pattern changes induced by *Utx* deficiency in brown fat. We found that a total of 1308 genes were differentially regulated by UTX deficiency (Log2 fold change ≥0.5 or ≤−0.5); out of which 254 genes were upregulated, and 1054 genes were downregulated by brown adipocyte UTX deficiency. As expected, bioinformatics analysis of these differentially expressed genes with an online software (https://github.com/PerocchiLab/ProFAT)[23] predicted an overall gene expression profile of reduced BAT characteristics in *Utx*-deficient iBAT, with a reciprocal increase in gene expression profile resembling that of WAT (Fig. 2A). This was consistent with a downregulation of brown fat-specific gene expression in *Utx*-deficient iBAT (Fig. 2B). Surprisingly, a hierarchical cluster analysis disclosed a significant upregulation of myogenic marker genes in *Utx*-deficient brown fat (Fig. 2C). Further analysis with quantitative RT-

PCR confirmed that myogenic marker gene expression was significantly upregulated in iBAT of chow-Fed, HFD-fed or cold-challenged UTXKO mice, including skeletal muscle myosin heavy polypeptide 1 (*Myh1*), skeletal muscle myosin heavy polypeptide 4 (*Myh4*), sarco/endoplasmic reticulum Ca$^{2+}$-ATPase isoform 1/ ATPase, Ca$^{2+}$ transporting, cardiac muscle, fast twitch 1 (*Serca1/ Atp2a1*), skeletal muscle α1 actin (*Acta1*), myogenic differentiation 1 (*Myod1*), myogenin (*Myog*) (Fig. 2D–F) and *Ckm* (Supplementary Fig. 5C–E). This was consistent with IHC analysis showing upregulation of skeletal muscle marker myosin heavy chain (MyHC) in iBAT of UTXKO mice (Fig. 2G). In addition, primary brown adipocytes isolated from *Utx*-deficient iBAT exhibited reduced oxygen consumption rate (OCR) as measured by Seahorse analyzer (Fig. 2H), indicating impaired mitochondrial function. To investigate whether UTX regulates BAT-to-myocyte remodeling via a cell-autonomous manner, we knocked down *Utx* in a brown adipocyte cell line BAT1 cells[21,24]. As expected, knocking down *Utx* in BAT1 cells significantly upregulated MyHC immunostaining (Fig. 2I), indicating UTX's regulation of BAT-to-myocyte remodeling is mediated via a cell-autonomous manner. Thus, our data revealed an intriguing BAT-to-myocyte remodeling process in BAT of UTXKO mice, which may lead to impaired brown adipocyte mitochondrial function and thermogenesis, thereby contributing to reduced energy expenditure and obesity in UTXKO mice.

Cold and diet are two primary triggers inducing brown fat thermogenesis[25–29]. Diet is a potent stimulator of energy expenditure, a phenomenon referred to as diet-induced thermogenesis that requires both brown fat and UCP1[25,26,28,29]. While cold-induced thermogenesis defends the body temperature of animals against a cold environment, diet-induced thermogenesis slows down weight gain induced by a short-term overfeeding[25,26]. We, therefore, explored whether brown fat myogenic remodeling could be modulated during cold- or diet-induced thermogenesis.

Interestingly, cold exposure, a well-known condition that boosts BAT thermogenic function, profoundly suppressed myogenic marker gene expression in iBAT (Supplementary Fig. 8). In addition, we also performed a time course study with chow- and HFD-fed animals to further investigate whether such brown adipocyte-myocyte remodeling process also occurred during the course of diet-induced obesity. As shown in Supplementary Fig. 9, HFD feeding induced *Ucp1* expression in iBAT starting from week 1 and lasted up to 12 weeks when compared to chow-fed animals, suggesting increased diet-induced thermogenesis (Supplementary Fig. 9A). However, the induction of *Ucp1* by HFD gradually disappeared such that *Ucp1* expression was no longer different between chow and HFD-fed animals after 24 weeks on HFD, suggesting that diet-induced thermogenesis waned after prolonged HFD feeding (Supplementary Fig. 9A). In contrast, HFD feeding suppressed the expression of myogenic marker genes including *Myod1*, *Myog* and *Myh1* at early stage of HFD feeding from week 1–12, while stimulating the expression of these myogenic marker genes after prolonged HFD feeding (Supplementary Fig. 9B–D). Further analysis revealed a reciprocal expression pattern of *Ucp1* and myogenic marker genes in iBAT of HFD-fed mice, showing a gradual decline of *Ucp1* expression and a reciprocal increase of myogenic marker gene expression following HFD feeding (Supplementary Fig. 9E). Indeed, the expression levels of myogenic marker genes were negatively correlated with that of *Ucp1* in iBAT when analyzed from both chow- and HFD-fed mice (Supplementary Fig. 9F).

Interestingly, we found that UTX protein level was significantly decreased in iBAT of mice fed HFD for 12 and 24 weeks (Fig. 3A). Further analysis also revealed a reciprocal pattern of UTX protein levels and myogenic marker gene expression in iBAT of HFD-fed mice, with a gradual decline of UTX protein

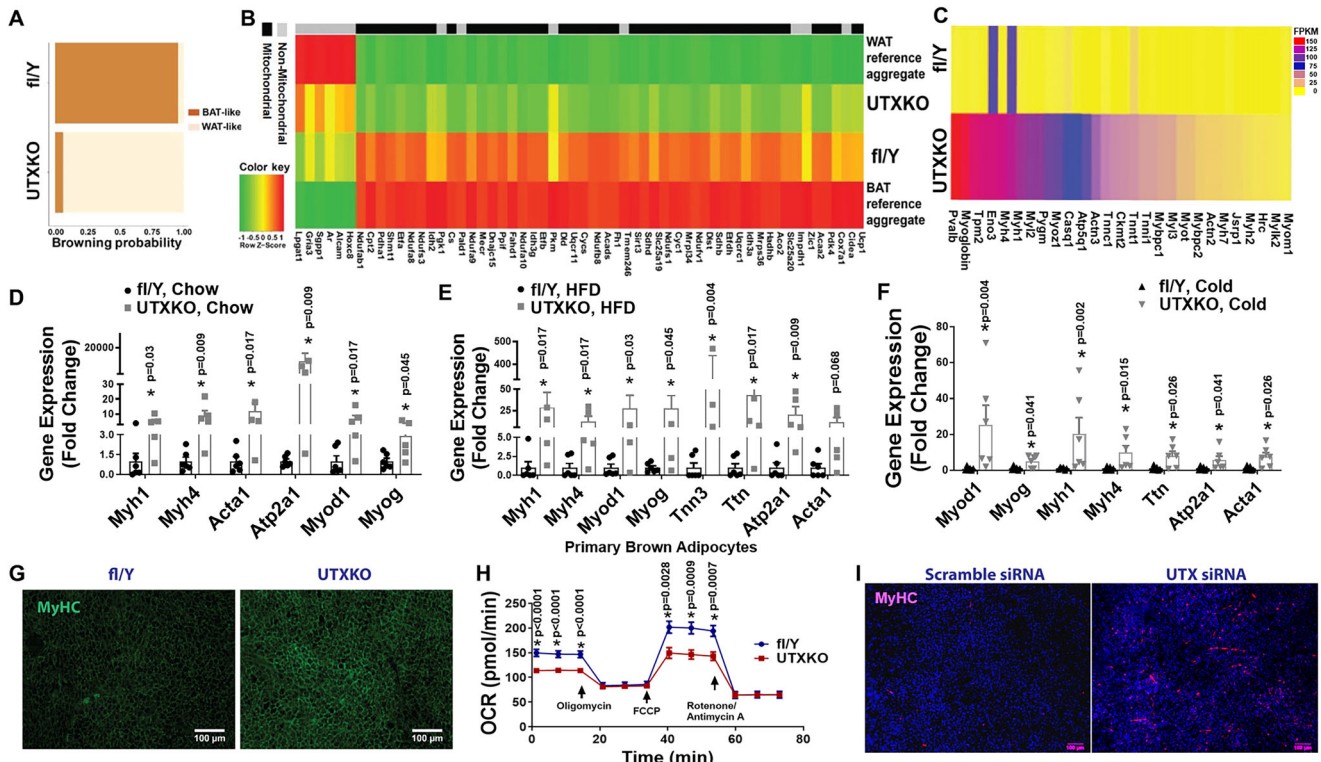

**Fig. 2 *Utx* deficiency in brown adipocytes induces BAT-to-myocyte remodeling. A** Bioinformatic modeling of BAT-like or WAT-like gene expression profiles using RNA-seq data from iBAT of male UTXKO and fl/Y mice fed HDF for 12 weeks using an online software (https://github.com/PerocchiLab/ProFAT). **B** RNAseq analysis of BAT-specific gene expression in iBAT of male UTXKO and fl/Y mice fed HDF for 12 weeks using an online software (https://github.com/PerocchiLab/ProFAT). The WAT reference aggregate and BAT reference aggregate were derived from the online software. **C** Heatmap of myogenic marker gene expression in iBAT of male UTXKO and fl/Y mice fed HDF for 12 weeks. **D–F** Quantitative RT-PCR analysis of myogenic marker gene expression in iBAT of chow-Fed (**D**, fl/Y = 6, UTXKO = 5), HFD-fed (**E**, fl/Y = 6, UTXKO = 5) UTXKO and fl/Y mice or in chow-fed 2-month-old male UTXKO and fl/Y mice after a 7-day cold challenge (n = 6/group) (F). *indicates statistical significance as marked in each panel between UTXKO and fl/Y by Mann–Whitney's nonparametric U test. **G** Representative immunohistochemistry (IHC) staining of myosin heavy chain (MyHC) in iBAT of UTXKO and fl/Y mice (n = 3 replicates for each group). **H** Oxygen consumption rate (OCR) in primary brown adipocytes isolated from iBAT of male UTXKO and fl/Y mice measured by a Seahorse XF 96 Extracellular Flux Analyzer (fl/Y = 17, UTXKO = 22). *Indicates statistical significance between UTXKO and fl/Y by two-tailed unpaired Student's t-test. **I** Representative immunohistochemistry (IHC) staining of myosin heavy chain (MyHC) in BAT1 brown adipocytes with *Utx* knockdown (n = 3 replicates for each group). All data are expressed as mean ± SEM.

level that corresponded to a reciprocal increase of myogenic marker gene expression following HFD feeding (Fig. 3B). An inverse correlation also existed between UTX protein levels and myogenic marker gene expression in iBAT when analyzed from HFD-fed mice (Fig. 3C).

To further confirm our findings, we performed RNA-seq analysis in iBAT of wild-type mice with chow or HFD feeding for 24 weeks. As expected, prolonged HFD feeding significantly induced myogenic marker gene expression in iBAT of HFD-fed mice compared to that of chow-fed mice (Fig. 3D). This was further confirmed by quantitative RT-PCR analysis showing upregulation of myogenic marker gene expression in iBAT of HFD-fed mice, including *Myh1*, *Myh4*, *Ckm*, *Myod1*, *Myog*, *Acta1*, *Atp2a1,* and Titin (*Ttn*) (Fig. 3E). In addition, IHC staining showed a significant upregulation of the skeletal muscle marker MyHC in iBAT of HFD-fed mice compared to that of chow-fed mice (Fig. 3F), further validating myogenic remodeling of iBAT under prolonged HFD feeding.

Recent data suggest that both stromal vascular fraction (SVF) cells and brown adipocytes contribute to *Myf5*[+] cell populations in lineage tracing studies[30]. In addition, it has been reported that brown preadipocytes express a prominent myogenic transcriptional signature, which is diminished upon brown adipocyte differentiation[31]. Thus, to investigate whether the observed

BAT-to-myocyte remodeling is derived from brown adipocytes or cells from SVF, we isolated primary brown adipocytes and SVF cells from iBAT of C57BL/6J mice fed chow or HFD for 24 weeks. Similar to previous report[31], we found that myogenic markers were expressed in both SVF cells and primary brown adipocytes; the expression of myogenic genes was relatively higher in SVF cells than that of primary brown adipocytes isolated from mice fed chow diet (Supplementary Fig. 10A). However, while HFD either did not change or only induced a mild increase in myogenic marker expression in SVF cells, HFD seemed to induce more profound myogenic marker expression in primary brown adipocytes than in SVF cells to a level similar to or higher than that of SVF levels (Supplementary Fig. 10A). The increase in myogenic gene expression in primary brown adipocytes was more evident when gene expression was normalized to that of chow values in each of the SVF and adipocyte groups (Supplementary Fig. 10B). Thus, our data suggest a prominent BAT-to-myocyte remodeling in primary brown adipocytes upon HFD feeding, which may primarily contribute to the BAT-to-myocyte switch observed in iBAT from HFD-fed mice. In consistence, primary brown adipocytes isolated from iBAT of 24-week HFD-fed mice exhibited reduced oxygen consumption rate (OCR) measured by Seahorse analyzer (Fig. 3G). Further, while acute HFD feeding significantly increased oxygen consumption in mice

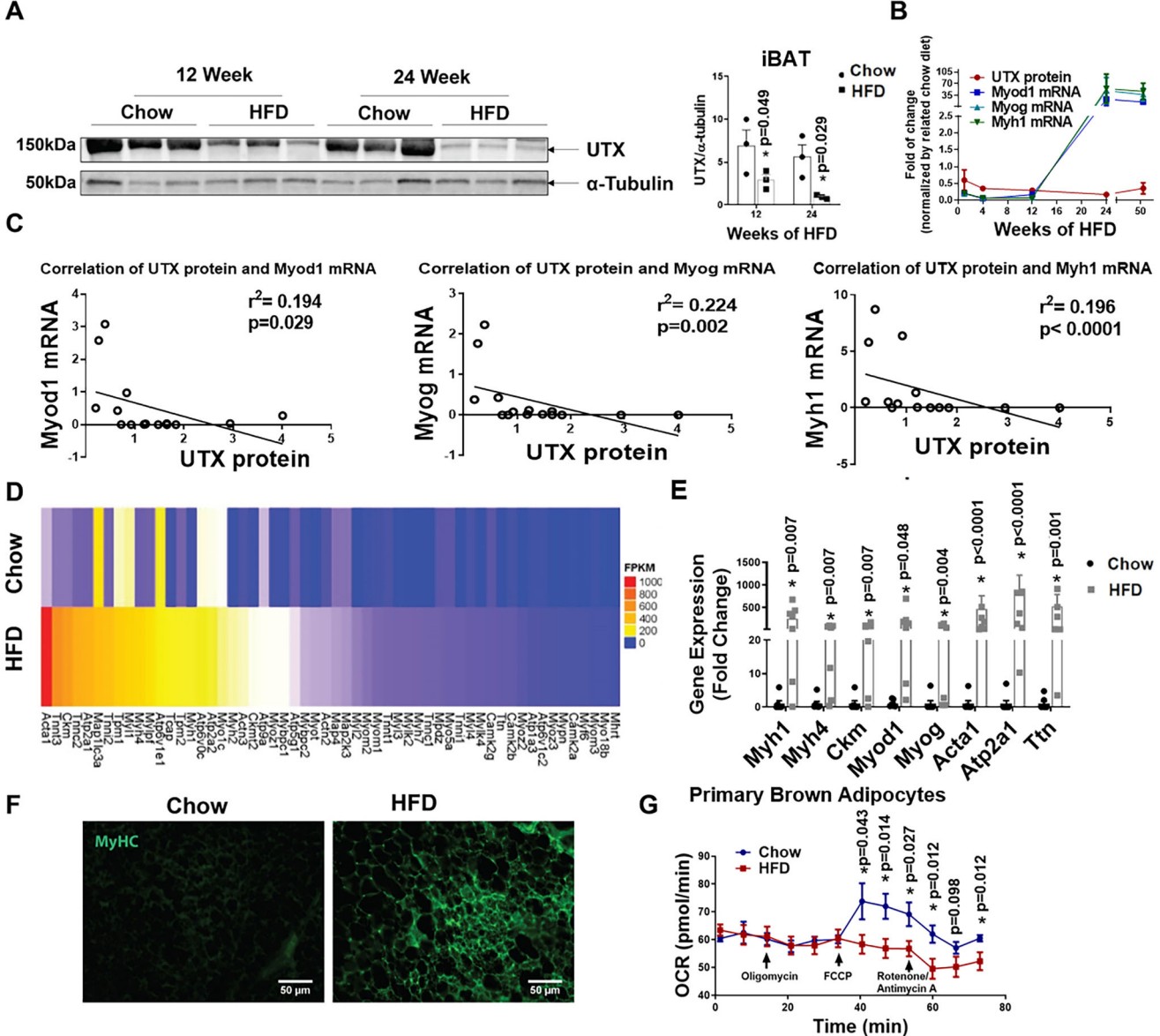

**Fig. 3 Long-term HFD feeding induces BAT-to-myocyte remodeling in iBAT. A** UTX protein level in iBAT of C57BL/6 J mice fed with HFD for 12 and 24 weeks ($n = 3$/group). *indicates statistical significance between chow and HFD by two-tailed unpaired Student's $t$-test. **B** Analysis of UTX protein levels and myogenic marker gene expression patterns in iBAT of HFD-fed mice for 1 week, 4 weeks, 12 weeks, 24 weeks, and 1 year. For UTX protein, $n = 3$/ group. For *Myod1*, *Myog,* and *Myh1* gene expression, $n = 8$/group. **C** Negative correlations between UTX protein levels and myogenic marker gene expression in iBAT of mice fed HFD for 1 week, 4 weeks, 12 weeks, 24 weeks, and 52 weeks ($n = 15$/group) as analyzed by Spearman's rank correlation coefficient test, $p = 0.029$ between UTX protein and *Myod1* mRNA, $p = 0.002$ between UTX protein and *Myog* mRNA, and $p < 0.0001$ between UTX protein and *Myh1* mRNA. **D** Heatmap of myogenic marker gene expression from iBAT of wild-type mice fed chow or HFD. **E** Quantitative RT-PCR analysis of myogenic marker gene expression in iBAT of chow- or HFD-fed wild-type C57BL/6J mice ($n = 7$/group). *Indicates statistical significance between Chow and HFD by Mann–Whitney's nonparametric $U$ test. **F** Representative IHC staining of MyHC in iBAT of chow- or HFD-fed wild-type C57BL/6J mice ($n = 3$ replicates for each group). **G** OCR of primary brown adipocytes isolated from chow- or HFD-fed wild-type C57BL/6J mice ($n = 24$ for chow, $n = 19$ for HFD). *Indicates statistical significance between chow and HFD by two-tailed unpaired Student's $t$-test. All data are expressed as mean ± SEM.

(Supplementary Fig. 11A), prolonged HFD feeding in mice resulted in a gradual reduction of oxygen consumption compared to that of chow-fed mice (Supplementary Fig. 11B–D).

To further study whether the BAT-to-myocyte remodeling process represents a common feature in obesity, we utilized *ob/ob* mice, where obesity is caused by genetic mutation at the leptin (*Lep*) gene[32]. We found that iBAT from *ob/ob* mice also exhibited significantly upregulated myogenic marker gene expression, including *Myod1*, *Myh1*, *Myh4*, *Myog*, *Acta1*, *Ckm,* and *Atp2a1* and significantly downregulated expression of BAT-specific

genes, including *Ucp1*, *Pgc1α*, *Ebf2*, *Cpt1b*, *Cox1*, acyl-Coenzyme A oxidase 1 (*Acox1*), *Pgc1β*, elongation of very-long-chain fatty acids (FEN1/Elo2, SUR4/Elo3, yeast)-like 3 (*Elovl3*), *Otop1* and epithelial V-like antigen 1 (*Eva1*) (Supplementary Fig. 12A–B).

Thus, our data revealed a reciprocal BAT-to-myocyte remodeling process in BAT that is modulated by both cold and diet challenges. The induction of this process in obesity impairs BAT mitochondria function and thermogenesis, which may contribute to reduced energy expenditure and the development of obesity.

The fact that HFD-induced obesity and brown adipocyte-specific *Utx* deletion induce a similar BAT-to-myocyte remodeling indicates that *Utx*-regulated epigenetic modification may be involved in this process.

**DNMT1 deficiency in brown fat induces myogenesis.** The interaction of epigenetic mechanisms, including histone methylation and DNA methylation, results in organization of the chromatin structure on different hierarchal levels, which coordinately regulates gene expression[33–35]. DNA methylation is catalyzed by DNA methyltransferases (DNMTs). While de novo DNA methylation is generally thought to be mediated by DNMT3A and DNMT3B, DNMT1 is believed to function as a maintenance enzyme to maintain DNA methylation status through mitosis using hemi-methylated DNA strands as templates[33–35]. However, recent data also suggest that DNMT1 may coordinate with DNMT3A and DNMT3B to regulate de novo DNA methylation[36–40]. Thus, to gain further insight into the epigenetic mechanisms that may regulate the BAT-myocyte remodeling process in brown adipocytes, we have generated mice with brown adipocyte-specific deletion of *Dnmt1*, *Dnmt3a* and *Dnmt3b* to study the role of DNA methylation in this process. Interestingly, we found that mice with brown adipocyte-specific deletion of *Dnmt1* or *Dnmt3a* exhibited similar phenotypes to that of UTXKO (see the results described below), whereas mice with brown adipocyte-specific deletion of Dnmt3b displayed a different metabolic phenotype (Xue and Shi, unpublished data). Thus, we have focused on brown adipocyte DNMT1 and DNMT3A in the current study and further explored whether UTX interacts with DNMTs in the regulation of BAT-myocyte remodeling in brown fat.

To generate mice with brown fat-specific deletion of *Dnmt1* (D1KO), we crossed *Dnmt1*-floxed mice (fl/fl)[41] with *Ucp1-Cre* mice[16]. We found *Dnmt1* deletion in brown fat resulted in around 50% reduction of *Dnmt1* mRNA expression (Supplementary Fig. 13). Interestingly, RNA-seq analysis using iBAT from D1KO and fl/fl mice revealed a significant upregulation of myogenic gene expression in *Dnmt1*-deficient brown fat (Fig. 4A), similarly to that observed in iBAT of HFD-fed, *ob/ob* and UTXKO mice. This was further confirmed by quantitative RT-PCR analysis showing upregulation of key myogenic gene expression, including *Myod1*, *Myog*, *Myh1*, *Myh4*, *Ckm*, *Serca1/Atp2a1*, *Ttn*, and *Acta1* in iBAT of D1KO mice (Fig. 4B). More importantly, bioinformatic analysis of RNAseq data from iBAT of HFD-fed, UTXKO and D1KO revealed a group of myogenic marker genes that were similarly upregulated in all three datasets (Fig. 4C), indicating that both DNMT1 and UTX might be involved in the regulation of BAT-to-myocyte remodeling in iBAT of HFD-fed mice.

To better label UCP1+ brown adipocytes, we generated mice with brown adipocytes-specific expression of GFP by triple-crossing *Dnmt1*-floxed mice, *Rosa-Gfp* mice[42], and *Ucp1*-Cre mice (D1KO-GFP, or *Ucp1-Cre::Dnmt1^{fl/fl}: Rosa-Gfp^{fl/fl}*). IHC staining indicated that *Dnmt1* deficiency markedly induced skeletal muscle marker MyHC in GFP-labeled brown adipocytes in D1KO-GFP mice (Fig. 4D). In addition, there was a reciprocal downregulation of brown fat thermogenic gene expression in iBAT from D1KO mice (Fig. 4E), which was associated with decreased UCP1 protein content and UCP1 IHC staining in iBAT of D1KO mice (Fig. 4F, G). Similar to that of *Utx*-deficient iBAT, the BAT-to-myocyte remodeling in *Dnmt1*-deficient primary brown adipocytes resulted in reduced OCR (Fig. 4H), indicative of impaired mitochondrial function. On the other hand, bioinformatic analysis of RNA-seq data with an online software[23] predicted that iBAT from D1KO mice exhibited gene expression

profiles characteristic of that of white fat (Fig. 4I), with less expression of BAT- and mitochondria-enriched genes (Fig. 4J).

To further study whether *Dnmt1* regulated BAT-to-myocyte remodeling via a cell-autonomous manner, we knocked down *Dnmt1* in 4-day differentiated BAT1 brown preadipocytes via siRNA approach. As expected, knockdown of *Dnmt1* in BAT1 cells significantly promoted myogenic gene expression, including *Myog*, *Myod1*, *Ckm*, *Myh1*, *Acta1*, *Atp2a1*, and *Ttn* (Fig. 5A) while simultaneously downregulated BAT-specific gene expression, including *Ucp1*, *Pgc1α*, *Ebf2*, and *Ebf3* (Fig. 5B). Moreover, knockdown of *Dnmt1* enhanced myogenesis as evidenced by increased MyHC staining (Fig. 5C), while simultaneously reducing brown adipogenesis as evidenced by much less neutral lipid accumulation measured by Oil red O staining (Fig. 5D). In consistence, Seahorse analysis revealed a decreased OCR in *Dnmt1* knockdown BAT1 brown adipocytes and similarly but to a greater extent, in *Utx* knockdown BAT1 brown adipocytes (Fig. 5E).

**Mice with Dnmt1 deficiency in brown fat exhibit increased body weight and adiposity on chow diet and are susceptible to DIO when fed a HFD.** To determine the role of brown adipocyte DNMT1 in the regulation of body weight and energy metabolism in animal models, we conducted metabolic characterization in D1KO mice. As shown in Fig. 5F, female D1KO mice gained more weight even on a regular chow diet compared to their fl/fl littermate control mice. In consistence, D1KO mice exhibited a significant increase in fat mass and a significant decrease in lean mass (Fig. 5G). Individual fat pad mass including iWAT, gonadal WAT (gWAT) and retroperitoneal WAT (rWAT) was also increased in D1KO compared to that of fl/fl mice (Fig. 5H). In addition, histological examination revealed larger adipocyte size in iBAT, iWAT, and gWAT of female D1KO mice (Fig. 5I). Using PhenoMaster metabolic cage systems, we found that female D1KO mice displayed lower energy expenditure with decreased oxygen consumption (Fig. 5J and Supplementary Fig. 14A), while there was no change in RER, locomotor activity, and a tendency of lower food intake in D1KO (Supplementary Fig. 14B–D). Female D1KO mice also displayed insulin resistance on the regular chow diet, as shown by increased fasting and fed insulin levels (Fig. 5K) and impaired glucose tolerance and insulin sensitivity in GTT and ITT tests (Fig. 5L–M).

We also investigated the involvement of UCP1-independent thermogenic pathways, including the creatine cycling and calcium futile cycle, as well as BAT-secreted batokines in regulating obesity-prone phenotypes in D1KO mice. We found that the expression of *Ckm* was significantly increased in iBAT of chow-fed female D1KO mice, along with the increase of other myogenic markers (Fig. 4B), whereas the expression of *Atp2a2* was not altered in iBAT of chow-fed female D1KO mice (Supplementary Fig. 15A). In addition, the expression of batokines *Nrg4*, *Pm20d1* and *Mstn* was also not altered in iBAT of chow-fed female D1KO mice (Supplementary Fig. 15B–C).

Similar results were also observed in male D1KO mice. When fed a regular chow diet, male D1KO mice also exhibited increased body weight and adiposity compared to their fl/fl littermate control mice (Supplementary Fig. 16A–B). They also displayed glucose intolerance and insulin resistance in GTT and ITT tests compared to fl/fl mice (Supplementary Fig. 16C–D).

To determine the role of brown adipocyte DNMT1 in diet-induced obesity, we challenged male and female D1KO mice and their littermate control fl/fl mice with HFD for up to 24 weeks. As expected, HFD-fed female D1KO mice gained more weight compared to fl/fl mice (Supplementary Fig. 17A). Body composition analysis revealed a significant increase in fat mass

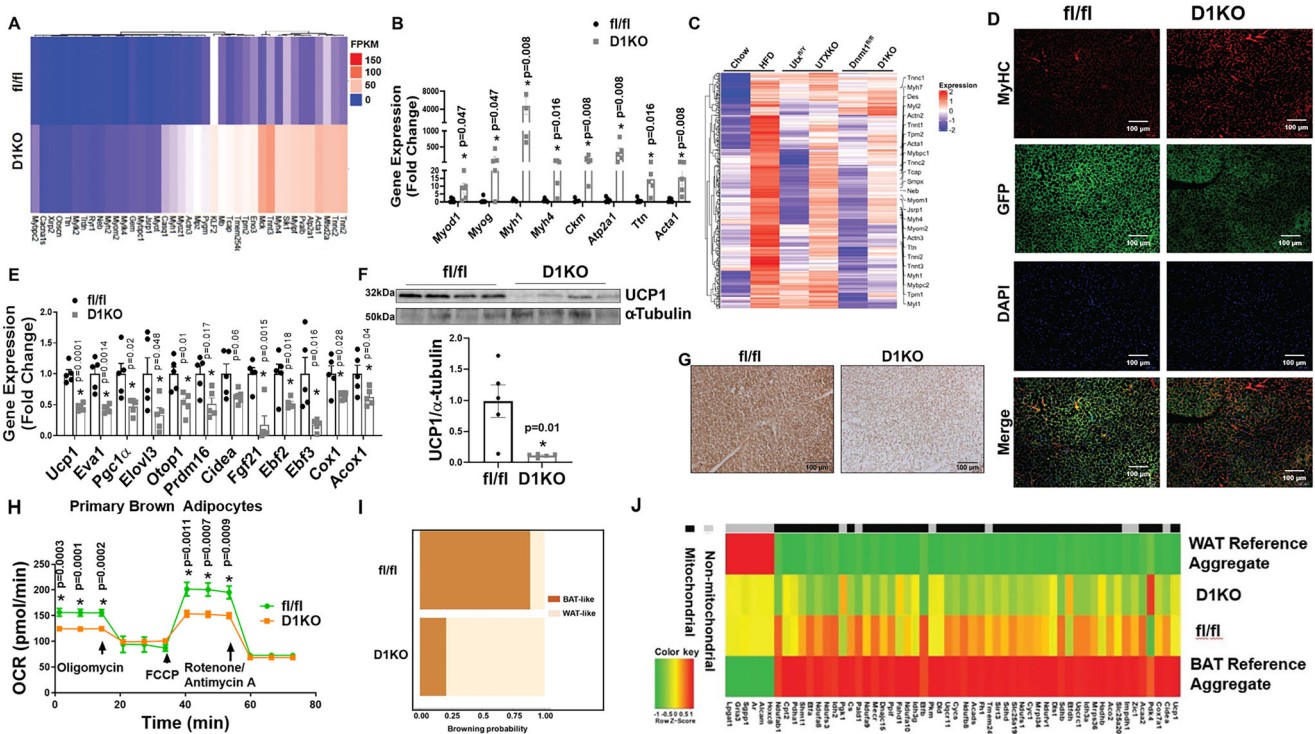

**Fig. 4 Dnmt1 deficiency in brown fat induces BAT-to-myocyte remodeling. A** Heatmap of myogenic gene expression in iBAT from D1KO and fl/fl mice on regular chow diet. **B** Quantitative RT PCR analysis of myogenic gene expression in D1KO and fl/fl mice on chow diet (n = 5/group). *Indicates statistical significance between fl/fl and D1KO by Mann–Whitney's nonparametric U test. **C** Hierarchical cluster analysis of genes similarly upregulated in iBAT of HFD-fed, UTXKO and D1KO mice. **D** Representative IHC staining of MyHC in GFP-labeled brown adipocytes in iBAT of D1KO-GFP and fl/fl-GFP mice on chow diet (n = 3 replicates per group). **E** Quantitative RT-PCR analysis of BAT-specific gene expression in iBAT of D1KO and fl/fl mice on chow diet (n = 5/group). *Indicates statistical significance between fl/fl and D1KO by unpaired two-tailed Student's t-test. **F** Immunoblotting analysis of UCP1 protein levels in iBAT of D1KO and fl/fl mice on chow diet (n = 5/group). *Indicates statistical significance between fl/fl and D1KO by unpaired two-tailed Student's t-test. **G** Representative IHC staining of UCP1 in iBAT of D1KO and fl/fl mice on chow diet (n = 3 replicates per group). **H** OCR of primary brown adipocytes isolated from iBAT of D1KO or fl/fl mice measured by a Seahorse analyzer (n = 14 for fl/fl, 20 for D1KO). *Indicates statistical significance between fl/fl and D1KO by unpaired two-tailed Student's t-test. **I** Bioinformatic modeling of BAT-like or WAT-like gene expression profiles using RNA-seq data from iBAT of D1KO and fl/fl mice on chow diet using an online software (https://github.com/PerocchiLab/ProFAT). **J** BAT-specific gene expression in iBAT of D1KO and fl/fl mice on chow diet using an online software (https://github.com/PerocchiLab/ProFAT). The WAT reference aggregate and BAT reference aggregate were derived from the online software. All data are expressed as mean ± SEM.

and a significant decrease in lean mass (Supplementary Fig. 17B), which was associated with increased individual fat pad mass in iWAT, gWAT, and iBAT (Supplementary Fig. 17C). Female D1KO mice on HFD also exhibited reduced energy expenditure with decreased oxygen consumption (Supplementary Fig. 17D–E) without changes in RER, locomotor activity, and food intake (Supplementary Fig. 17F–H). With the increased adiposity, D1KO mice had increased fed insulin levels (Supplementary Fig. 17I) and displayed glucose intolerance and insulin resistance as assessed by GTT and ITT (Supplementary Fig. 17J–K), respectively.

Similarly, male D1KO mice gained more weight compared to their fl/fl littermate control mice upon HFD challenge (Supplementary Fig. 18A), which was associated with increased fat mass in iWAT (Supplementary Fig. 18B) and larger adipocytes in iWAT and iBAT (Supplementary Fig. 18C). Male D1KO mice on HFD also exhibited lower energy expenditure with decreased oxygen consumption (Supplementary Fig. 18D–E) without changes in RER and locomotor activity (Supplementary Fig. 18F–G). In addition, male D1KO mice had slightly decreased cumulative food intake (Supplementary Fig. 18H), possibly due to adaptation to their reduced energy expenditure. Further, male D1KO mice exhibited downregulated thermogenic gene expression in iBAT and iWAT (Supplementary Fig. 18I–J), with decreased UCP1 protein levels in

iBAT (Supplementary Fig. 18K). Consistent with their increased adiposity, male D1KO mice displayed glucose intolerance and insulin resistance as assessed by GTT and ITT (Supplementary Fig. 18L–M). Thus, our data consistently indicate that DNMT1 deficiency in brown fat promotes obesity and impairs insulin sensitivity in mice.

**DNMT1 deficiency in brown fat impairs cold-induced thermogenesis in iBAT**. To study the role of brown fat DNMT1 in cold-induced thermogenesis, we challenged chow-fed 2-month-old D1KO and fl/fl mice for either an acute (4-h) or a chronic (7-day) cold exposure at 5 °C. Interestingly, D1KO mice exhibited lower body temperature during acute cold exposure (Fig. 6A), with a bigger temperature reduction than control fl/fl mice within the first 2 h of cold exposure (Fig. 6B), suggesting that the DNMT1-deficient mice are cold sensitive. During the acute cold exposure, Dnmt1 deficiency in iBAT resulted in reduced BAT-specific gene expression, with simultaneously upregulated myo-genic marker gene expression (Fig. 6C, D). Similarly, during the chronic cold exposure, DNMT1 deficiency in iBAT also resulted in reduced expression of BAT-specific genes, while reciprocally increasing the expression of myogenic marker genes in iBAT of D1KO mice (Fig. 6E, F). In consistence with gene expression profiles, iBAT from D1KO mice also exhibited decreased UCP1

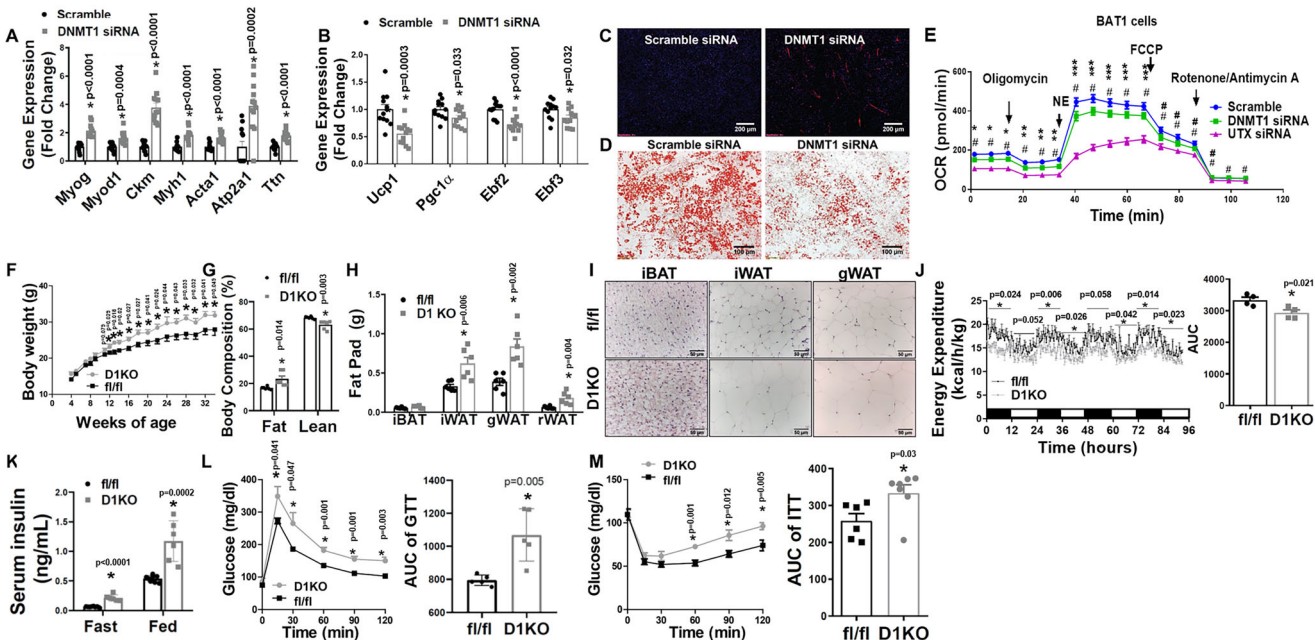

**Fig. 5 DNMT1 deficiency in brown fat impairs brown fat function. A, B** Quantitative RT-PCR analysis of myogenic gene expression (**A**, $n = 12$/group) and BAT-specific gene expression (**B**, $n = 12$/group) in BAT1 brown adipocytes transfected with scramble or *Dnmt1* siRNA. *Indicates statistical significance between Scramble and DNMT1 siRNA groups by two-tailed unpaired Student's *t*-test. **C, D** Representative IHC staining of MyHC (**C**) and Oil red O staining (**D**) in BAT1 brown adipocytes transfected with scramble or *Dnmt1* siRNA ($n = 3$ replicates for each group). **E** OCR in BAT1 brown adipocytes transfected with scramble, *Dnmt1* or *Utx* siRNA measured by a Seahorse analyzer. $n = 24$ for Scramble, 31 for DNMT siRNA, and 24 for UTX siRNA; statistical significance was analyzed by one-way ANOVA with repeated measure followed by Fisher's Least Significant Difference (LSD) test, $F (2, 76) = 35.433$, $p < 0.0001$. *$p < 0.01$, **$p < 0.0001$, ***$p < 0.05$ between Scramble and DNMT1 siRNA; #$p < 0.0001$, ##$p < 0.01$ between Scramble and UTX siRNA. For (**A**–**E**), BAT1 cells were differentiated into brown adipocytes as described under Methods and scramble and targeting siRNAs were transfected into day 4 differentiated BAT1 cells using Amaxa Nucleofector II Electroporator with an Amaxa cell line nucleofector kit L according to the manufacturer's instructions (Lonza). Cells were harvested 2 days after for further analysis. **F–H** Body weight growth curve (**F**, $n = 8$ for fl/fl and 6 for D1KO), Body composition (**G**, $n = 6$/group), and Fat pad weight (**H**, $n = 6$/group) in female D1KO and fl/fl mice on chow diet. Female D1KO and fl/fl mice were weaned onto regular chow diet and various metabolic phenotypes were studied. *Indicates statistical significance between fl/fl and D1KO by two-tailed unpaired Student's *t*-test. **I** Representative H&E staining of iWAT, iBAT and gWAT of female D1KO and flf/fl mice on chow diet ($n = 4$ replicates per group). **J** Energy expenditure in female D1KO and fl/fl mice on chow diet ($n = 4$/group). Left, *indicates statistical significance between fl/fl and D1KO analyzed by ANOVA with repeated measures: time 0–12 h, $F(1,6) = 8.962$, $p = 0.024$; time 12–24 h, $F(1,6) = 5.819$, $p = 0.052$; time 24–36 h, $F(1,6) = 16.794$, $p = 0.006$; time 36–48 h, $F(1,6) = 8.664$, $p = 0.026$; time 48–60 h, $F(1,6) = 5.457$, $p = 0.058$; time 60–72 h, $F(1,6) = 6.637$, $p = 0.042$; time 72–84 h, $F(1,6) = 11.648$, $p = 0.014$; time 84–96 h, $F(1,6) = 9.294$, $p = 0.023$. Right, *indicates statistical significance between fl/fl and D1KO by two-tailed unpaired Student's *t*-test. **K–M** Fasting and fed serum insulin levels (**K**, $n = 8$ for fl/fl and 6 for D1KO), GTT (**L**, $n = 5$/group), and ITT (**M**, $n = 6$ for fl/fl and 7 for D1KO) of female D1KO and fl/fl mice on chow diet. *Indicates statistical significance between fl/fl and D1KO by two-tailed unpaired Student's *t*-test. All data are expressed as mean ± SEM.

protein level (Fig. 6G), larger brown adipocytes as measured by H&E staining (Fig. 6H), and less UCP1 IHC staining in response to the chronic cold exposure (Fig. 6I). These data suggest that DNMT1 deficiency in brown fat impairs cold-induced thermogenesis.

We also studied whether deleting *Dnmt1* in brown adipocytes modulates beige adipocyte induction in WAT. Interestingly, in contrast to impaired thermogenesis in iBAT of cold-challenged D1KO mice (Fig. 6), the expression of *Ucp1* and other thermogenic genes were either tended to increase or moderately increase in iWAT of cold-challenged D1KO mice (Supplementary Fig. 19A). In addition, UCP1-immunostaining also clearly showed an increase in beige adipocyte formation in iWAT of D1KO mice compared to fl/fl littermates after a 7-day cold exposure (Supplementary Fig. 19B). Similarly, thermogenic gene expression and beige adipocyte formation were also increased in eWAT of D1KO compared to that of fl/fl littermates after the chronic cold exposure (Supplementary Fig. 19C–D). These data indicate an intriguing dissociation of thermogenesis between traditional brown adipocytes in iBAT and beige adipocytes identified in WAT. Recent data suggest that BAT and WAT are

derived from different developmental origins. Brown adipocytes from iBAT and skeletal muscle cells share a common developmental lineage and are derived from precursor cells that express *Myf5*, whereas most white adipocytes from eWAT and iWAT come from a different lineage origin that does not express *Myf5*[43,44]. Thus, it is possible that the role of DNMT1 in the determination of BAT-muscle switch may be specific to the *Myf5*-positive lineage cells in iBAT, but not the *My5*-negative lineage cells in eWAT and iWAT. The increased beiging in iWAT and eWAT of D1KO mice in response to cold is possibly a compensatory adaptation to reduced thermogenic function in iBAT.

**Mice with brown adipocyte-specific Dnmt3a deficiency displayed similar metabolic phenotypes to that of UTXKO and D1KO.** We also generated mice with brown fat-specific deletion of *Dnmt3a*, a DNA methyltransferase that is responsible for de novo DNA methylation (D3aKO), by crossing *Dnmt3a*-floxed mice[45] with *Ucp1-Cre* mice[16], and found that D3aKO mice had similar metabolic phenotypes to that of D1KO. For instance,

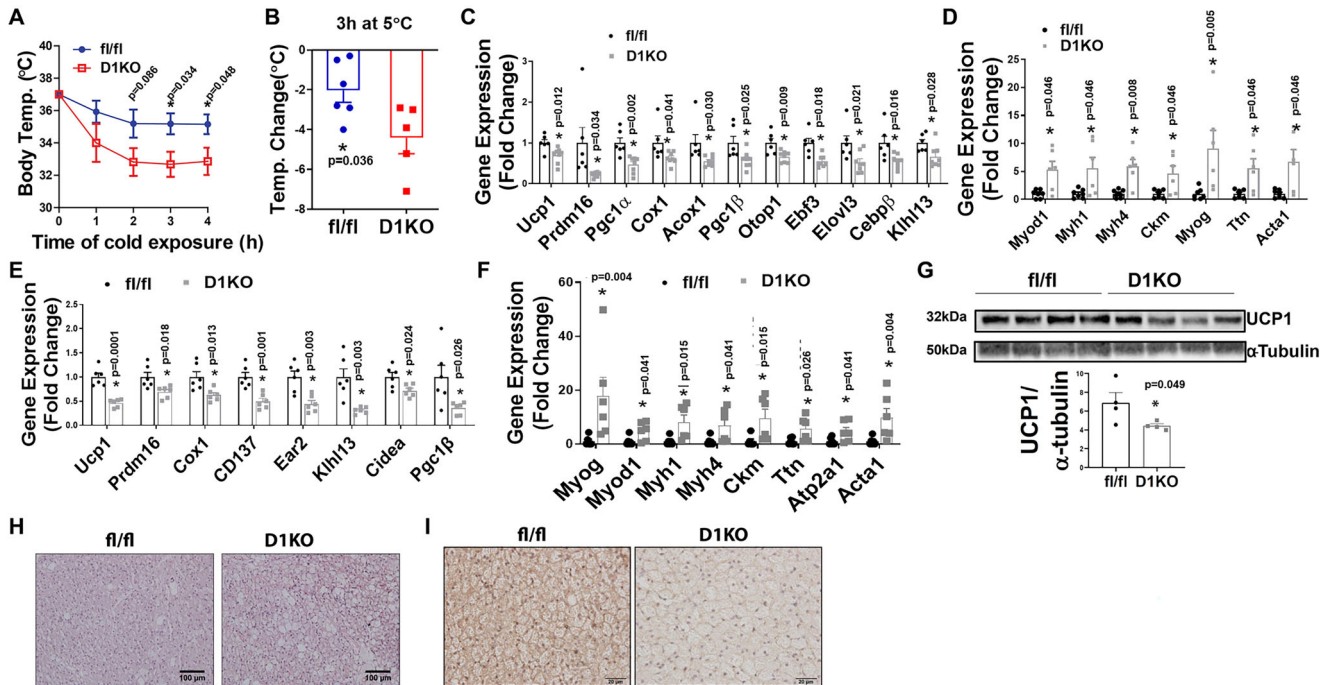

**Fig. 6 DNMT1 deficiency in brown fat impairs cold-induced thermogenesis. A** Body temperature in male chow-fed 2-month-old D1KO and fl/fl mice during acute 5 °C cold exposure ($n = 6$ for fl/fl and 5 for D1KO). *Indicates statistical significance between fl/fl and D1KO as analyzed by two-tailed unpaired Student's $t$-test. **B** Body temperature change in male chow-fed 2-month-old D1KO and fl/fl mice after 3 h of cold exposure ($n = 6$ for fl/fl and 5 for D1KO). *Indicates statistical significance between fl/fl and D1KO as analyzed by two-tailed unpaired Student's $t$-test. **C, D** Quantitative RT-PCR analysis of BAT-specific gene expression (**C**, $n = 6$ for fl/fl and 8 for D1KO), and myogenic gene expression (**D**, $n = 7$ for fl/fl and 6 for D1KO) in iBAT of male chow-fed 2-month-old D1KO and fl/fl mice after an acute 4-h 5 °C cold exposure. *Indicates statistical significance between fl/fl and D1KO as analyzed by two-tailed unpaired Student's $t$-test in (**C**) and Mann–Whitney's nonparametric $U$ test in (**D**). **E, F** Quantitative RT-PCR analysis of BAT-specific gene expression (**E**, $n = 6$/group), and myogenic gene expression (**F**, $n = 6$/group) in iBAT of male chow-fed 2-month-old D1KO and fl/fl mice after a chronic 7-day 5 °C cold exposure. *Indicates statistical significance between fl/fl and D1KO as analyzed by two-tailed unpaired Student's $t$-test in (**E**) and Mann–Whitney's nonparametric $U$ test in (**F**). **G** Immunoblotting of UCP1 protein in iBAT of male chow-fed 2-month-old D1KO and fl/fl mice after a 7-day 5 °C cold exposure ($n = 4$/group). *Indicates statistical significance between fl/fl and D1KO as analyzed by two-tailed unpaired Student's $t$-test. **H, I** Representative H&E staine (**H**) and UCP1 IHC staining (**I**) in iBAT from male chow-fed 2-month-old D1KO and fl/fl mice after a chronic 5 °C 7-day cold exposure ($n = 3$ replicates for each group). All data are expressed as mean ± SEM.

female D3aKO mice gained more weight even on regular chow diet compared to their fl/fl littermate control mice (Supplementary Fig. 20A) even with less food intake (Supplementary Fig. 20B). In consistence, D3aKO mice exhibited larger fat pads and liver compared to that of fl/fl mice (Supplementary Fig. 20C). Using PhenoMaster metabolic cage systems, we found that D3aKO mice displayed lower energy expenditure (Supplementary Fig. 20D) with decreased oxygen consumption (Supplementary Fig. 20E) with largely no changes in locomotor activity (Supplementary Fig. 20F). Further GTTs and ITTs confirmed that D1KO mice developed glucose intolerance and insulin resistance (Supplementary Fig. 20G–H).

**Myod1 mediates the effect of DNMT1 deficiency on brown fat myogenesis.** To further determine potential myogenic transcriptional factors whose upregulations are direct targets by promoter demethylation due to *Dnmt1* deficiency, we performed a DNA methylation profiling experiment using Reduced Representation Bisulfite Sequencing (RRBS) approach[46–48]. Since our data suggest that iBAT from female D1KO mice exhibits prominent BAT-to-myocyte remodeling compared to their fl/fl littermates even on a regular chow diet, we used DNA samples from iBAT of chow-fed female D1KO and fl/fl mice for the RRBS analysis. We found there are up to 242 differentially methylated regions (DMRs) in iBAT of D1KO vs fl/fl mice. Within these DMRs, around 202 were located within genes, with the rest

associated with gene promoters. Notably, our RRBS profiling indicated that the methylation rate at the proximate promoter and 5'-region of *Myod1* gene was significantly decreased in iBAT of D1KO mice compared to that of fl/fl mice (Fig. 7A), whereas *Myod1* express was reciprocally upregulated in iBAT from UTXKO, HFD-fed and D1KO mice (Figs. 2–4, and Supplementary Fig. 9). Interestingly, the proximal promoter and 5' region of *Myod1* are enriched with CpG sites (Supplementary Fig. 21A); alterations of DNA methylation on these CpG sites could potentially alter *Myod1* expression.

Prior lineage tracing studies show that brown fat and skeletal muscle indeed share the same developmental origins[44]. To determine the role of *Myod1* in the determination of brown fat and myocyte lineage, we interrogated the relationship between the myogenic driver *Myod1* and the BAT marker *Ucp1* by measuring the expression pattern of these two genes during brown fat development. *Ucp1* expression in iBAT began to increase at embryonic day 17 (E17), and continued to rise postnatally (Fig. 7B), similar to previous reports[49–51]. In contrast, *Myod1* expression was at the highest level in iBAT at E17, which then sharply declined afterwards and stayed at low levels postnatally (Fig. 7C). These data suggest that *Myod1* and *Ucp1* expression is inversely correlated and may be mutually exclusive. Moreover, reciprocal regulation of *Myod1* and *Ucp1* expression was also evident in cold exposure such that cold exposure markedly stimulated *Ucp1* expression (Fig. 7D) within 24 h while

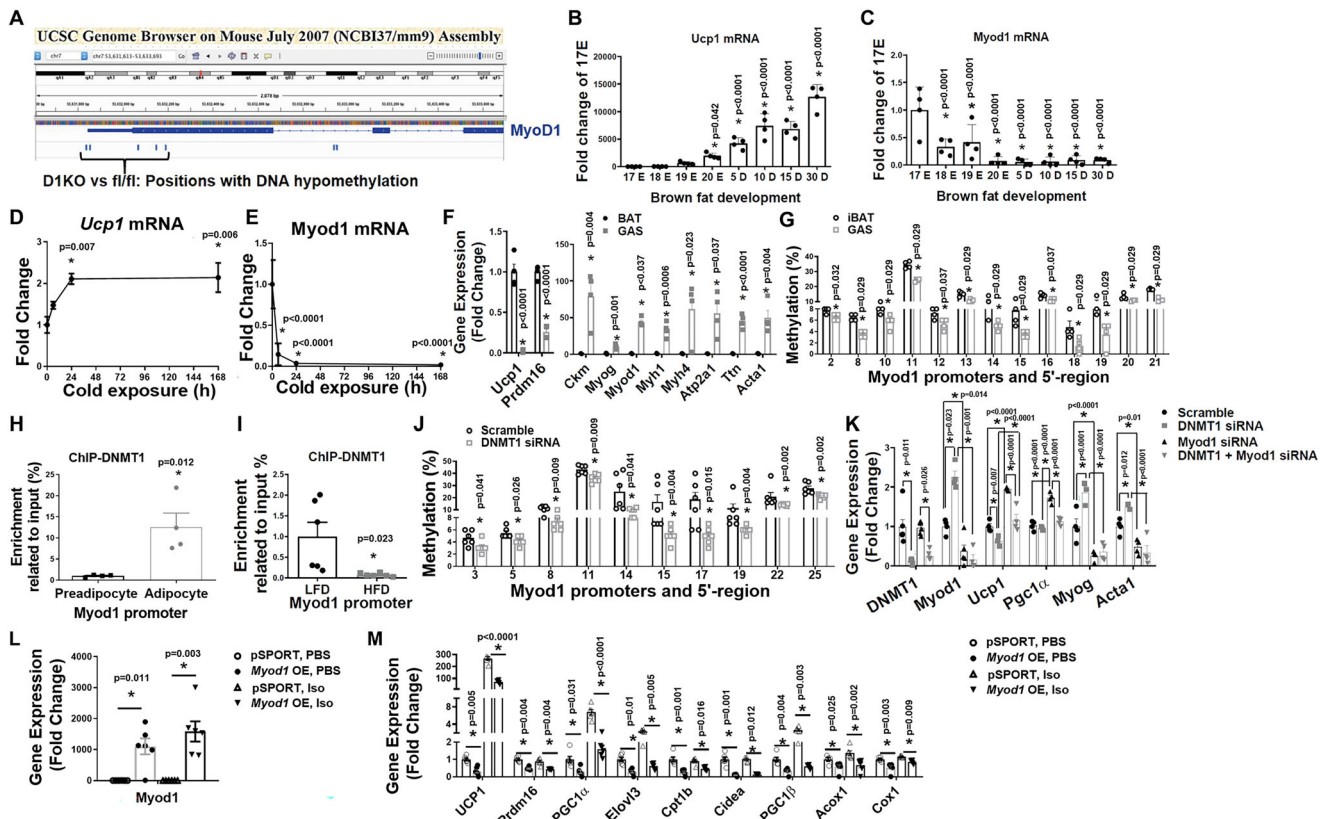

**Fig. 7 MyoD1 mediates the effect of DNMT1 deficiency on brown fat myogenesis. A** RRBS profiling of DNA methylation level at MyoD1 promoter in iBAT of D1KO and fl/fl mice. **B, C** *Ucp1* (**B**, $n = 4$/group) and *Myod1* (**C**, $n = 4$/group) expression in iBAT of mice during late embryonic and postnatal development. *Indicates statistical significance vs. 17E with one-way ANOVA followed by Fisher's LSD multiple comparisons test; in (**B**), $F_{(7,24)} = 48.31$, $p < 0.0001$, and in (**C**), $F_{(7,24)} = 10.54$, $p < 0.0001$. **D, E** *Ucp1* (**D**) and *Myod1* (**E**) expression in iBAT of mice during cold exposure ($n = 3$/group). *indicates statistical significance vs. Time 0 with one-way ANOVA followed by Fisher's LSD multiple comparisons test; in (**D**), $F_{(3,8)} = 6.406$, $p = 0.016$, and in (**E**), $F_{(3,8)} = 25.096$, $p < 0.0001$. **F** *Ucp1*, *Prdm16* and myogenic marker gene expression in iBAT and gastrocnemius (GAS) muscle ($n = 4$/group). *Indicates statistical significance between iBAT and GAS as analyzed by two-tailed unpaired Student's *t*-test, except for *Myod1* and *Atp2a1*, which were analyzed by Mann–Whitney's nonparametric *U* test. **G** Pyrosequencing analysis of DNA methylation level at *Myod1* promoter in iBAT and GAS muscle ($n = 4$/group). *Indicates statistical significance between iBAT and GAS as analyzed by Mann–Whitney's nonparametric *U* test. **H** ChIP assay of DNMT1 binding to *Myod1* promoter in undifferentiated BAT1 preadipocytes and differentiated BAT1 brown adipocytes ($n = 4$/group). *indicates statistical significance by two-tailed unpaired Student's *t*-test. **I** ChIP assay of DNMT1 binding to *Myod1* promoter in iBAT from HFD- or LFD-fed mice ($n = 6$/group). *Indicates statistical significance by two-tailed unpaired Student's *t*-test. **J** Pyrosequencing analysis of DNA methylation levels at *Myod1* promoter in BAT1 brown adipocytes transfected with scramble or *Dnmt1* siRNA ($n = 6$/group). *Indicates statistical significance between iBAT and GAS as analyzed by Mann–Whitney's nonparametric *U* test. **K** Quantitative RT-PCR analysis of myogenic marker gene and BAT gene expression in BAT1 brown adipocytes transfected with scramble, *Dnmt1*, *Myod1*, or *Dnmt1* + *Myod1* siRNA ($n = 4$/group). *Indicates statistical significance among groups. For Dnmt1 and Myod1, statistical significance was analyzed by Kruskal–Wallis non-parametric ANOVA H test by rank followed by Pairwise Comparisons test between groups, $H_{(3)} = 13.560$, $p = 0.004$ for Dnmt1, and $H_{(3)} = 13.097$, $p = 0.004$ for Myod1. For Ucp1, Pgc1α, Myog and Acta1, statistical significance was analyzed by one-way ANOVA followed by Fisher's LSD multiple comparisons test: for Ucp1, $F_{(3,12)} = 45.139$, $p < 0.0001$; for Pgc1α, $F_{(3,12)} = 51.81$, $p < 0.0001$; for Myog, $F_{(3,12)} = 33.178$, $p < 0.0001$; for Acta1, $F_{(3,12)} = 20.045$, $p < 0.0001$. **L, M** *Myod1* (**L**) and BAT-specific gene expression (**M**) in *Myod1*-overexpressed BAT1 brown adipocytes treated with PBS or isoproterenol (Iso). $n = 6$/group. *indicates statistical significance analyzed by Kruskal–Wallis non-parametric ANOVA H test by rank followed by Pairwise Comparisons test between groups. In (**L**), $H_{(3)} = 17.613$, $p = 0.001$. In (**M**), for Ucp1, $H_{(3)} = 21.6$, $p < 0.0001$; for Prdm16, $H_{(3)} = 17.553$, $p = 0.001$; for Pgc1α, $H_{(3)} = 20.309$, $p < 0.0001$; for Elovl3, $H_{(3)} = 19.62$, $p < 0.0001$; for Cpt1b, $H_{(3)} = 18.033$, $p < 0.0001$; for Cidea, $H_{(3)} = 18.023$, $p < 0.0001$; for pgc1β, $H_{(3)} = 21.367$, $p < 0.0001$; for Acox1, $H_{(3)} = 16.847$, $p = 0.001$; for Cox1, $H_{(3)} = 19.807$, $p = 0.0009$. For (**J–M**), BAT1 cells were differentiated into brown adipocytes as described under Methods. Scramble or targeting siRNAs, or control or *Myod1* overexpressing plasmids were transfected into day 4 differentiated BAT1 cells using Amaxa Nucleofector II Electroporator with an Amaxa cell line nucleofector kit L. Cells were harvested 2 days after for pyrosequencing or gene expression analysis. All data are expressed as mean ± SEM.

simultaneously down-regulating *Myod1* expression (Fig. 7E). The enrichment of myogenic genes such as *Myod1* along with other markers is a molecular signature of the skeletal muscle, which distinguishes gastrocnemius muscle from iBAT (Fig. 7F). Interestingly, the high expression of *Myod1* in skeletal muscle was associated with decreased methylation rate at a number of CpG sites at the *Myod1* promoter and 5′-region as measured by pyrosequencing analysis (Fig. 7G). Hypomethylation at the

*Myod1* promoter and 5′-region may contribute to higher *Myod1* gene expression in skeletal muscle, which may be important in maintaining myogenic signature in skeletal muscle.

We then further determined the role of DNMT1 in *Myod1* promoter methylation. Our ChIP assay revealed a significantly higher DNMT1 binding to the *Myod1* promoter in differentiated BAT1 brown adipocytes compared to undifferentiated BAT1 preadipocytes (Fig. 7H), indicating that enhanced methylation at

*Myod1* promoter due to DNMT1 binding may decrease *Myod1* expression in mature brown adipocytes. Further, downregulated binding of DNMT1 to *Myod1* promoter was also observed in iBAT from HFD-fed mice compared to LFD-fed mice (Fig. 7I). Decreased DNMT1 binding to *Myod1* promoter may increase its gene expression, resulting in the induction of myogenic program seen in the iBAT from HFD-fed mice (Fig. 3, and Supplementary Fig. 9). To determine whether the effect of DNMT1 on *Myod1* promoter methylation was via a cell-autonomous action, we knocked down *Dnmt1* in BAT1 cells. We found that *Dnmt1* knockdown in BAT1 brown adipocytes resulted in reduced methylation at a number of CpG sites at *Myod1* promoter (Fig. 7J).

To confirm our findings that *Myod1* serves as an epigenetic target for *Dnmt1* and mediates *Dnmt1* deletion-induced BAT-to-myocyte remodeling in brown adipocytes, we knocked down *Dnmt1* and *Myod1* individually or in combination in BAT1 brown adipocytes and measured BAT- and myogenic-specific gene expression. As expected, BAT1 brown adipocytes transfected with *Dnmt1* and/or *Myod1* siRNA had significantly reduced *Dnmt1* and/or *Myod1* expression, respectively, confirming the knockdown efficiency (Fig. 7K). Consistent with the decreased methylation at *Myod1* promoter by *Dnmt1* knockdown (Fig. 7J), *Dnmt1* knockdown in BAT1 brown adipocytes significantly upregulated *Myod1* expression, which was abolished by *Myod1* knockdown (Fig. 7K). In addition, *Dnmt1* knockdown in BAT1 brown adipocytes downregulated BAT-specific *Ucp1* expression while up-regulating myogenic gene expression, including *Myog* and *Acta1*, which was completely reversed by *Myod1* knockdown (Fig. 7K). Interestingly, we found *Myod1* knockdown in BAT1 cells significantly upregulated BAT-specific *Ucp1* and *Pgc1α* expression while down-regulating myogenic marker *Myog* and *Acta1* expression (Fig. 7K); whereas *Myod1* overexpression in BAT1 cells (Fig. 7L) was sufficient to suppress both basal- and the β-adrenergic agonist isoproterenol-stimulated thermogenic gene expression, including *Ucp1*, *Prdm16*, *Pgc1α*, *Elovl3*, *Cpt1β*, *Cidea*, *Pgc1β*, *Acox1* and *Cox1* (Fig. 7M). Thus, our in vitro data confirm our findings that DNMT1 determines BAT-myocyte remodeling via *Myod1* promoter methylation in brown adipocytes.

**Specifically reducing DNA methylation at Myod1 promoter induces myocyte-like brown adipocytes**. We next utilized a targeted demethylation approach to determine whether specifically targeting CpG sites and reducing their methylation status at *Myod1* promoter could recapitulate *Dnmt1* deletion-induced BAT-to-myocyte remodeling in brown adipocytes. To achieve site-specific demethylation at *Myod1* promoter, we employed a modified endonuclease dead version of CRISPR associated protein 9 (dCas9) fused with the catalytic domain (CD) of the enzyme involved in DNA demethylation, tet methylcytosine dioxygenase 1 (TET1) (dCas9-TET1CD)[52–56] (Supplementary Fig. 21B). We transfected BAT1 brown adipocytes with plasmids expressing dCas-TET1CD along with plasmids expressing either scramble non-targeting guide RNA (scramble-gRNA-mCherry) or gRNA targeting CpG sites at *Myod1* promoter (*Myod1*-gRNA-mCherry), and found that DNA methylation rate of a number of CpG sites at *Myod1* promoter was significantly reduced (Fig. 8A). The downregulated DNA methylation at *Myod1* promoter was associated with upregulation of *Myod1* mRNA expression and reciprocal downregulation of thermogenic gene expression such as *Ucp1*, *Pgc1α*, *Ebf2* and *Prdm16* (Fig. 8B), suggesting that we have successfully established an approach to specifically target DNA methylation at *Myod1* promoter.

To study the role of targeted DNA methylation at *Myod1* promoter in brown fat in vivo, we surgically injected lentiviruses expressing dCas9-TET1CD along with lentiviruses expressing either scramble-gRNA-mCherry or *Myod1*-gRNA-mCherry bilaterally into iBAT of male C57BL/6J mice on chow diet. IHC analysis with mCherry and perilipin antibodies confirmed successful infection of the lentiviruses into iBAT brown adipocytes (Supplementary Fig. 22). The mice receiving lentiviral injection of dCas9-TET1CD and *Myod1*-targeting gRNA exhibited higher body weight (Fig. 8C) and body fat composition (Fig. 8D) even on chow diet, thus mimicking the metabolic phenotypes of male and female D1KO mice on chow diet (Fig. 4 and Supplementary Fig. 16). Moreover, lentiviral treatment of dCas9-TET1CD significantly reduced DNA methylation rate of a number of CpG sites at *Myod1* promoter (Fig. 8E). The decreased methylation at *Myod1* promoter appeared to result in a marked induction of myogenic gene expression (Fig. 8F), which was associated with a downregulation of thermogenic gene expression (Fig. 8G). Further IHC staining revealed a significant upregulation of the skeletal muscle marker MyHC and a reciprocal downregulation of the thermogenic marker UCP1 (Fig. 8H). These data indicate that decreased DNA methylation at *Myod1* promoter initiates a BAT-to-myocyte remodeling process in brown fat.

**DNMT1 silences Myod1 expression via interacting with PRDM16**. We next determined the molecular mechanism underlying the strikingly similar BAT-to-myocyte remodeling phenotype in BAT of UTXKO and D1KO mice. Our previous data suggested that UTX promotes brown adipocyte thermogenic program via coordinated regulation of H3K27 demethylation and acetylation at *Pgc1α* promoter;[13] whereas PRDM16 is critical in controlling BAT-muscle switch[44]. We thus performed an assay for transposase-accessible chromatin sequencing (ATAC-seq) analysis in iBAT of UTXKO and fl/fl littermates fed HFD for 12 weeks. We compared the genome-wide alterations in chromatin accessibility landscape assessed by ATAC-seq with the corresponding gene expression assessed by RNA-seq and discovered a strong correlation between the chromatin accessibility status and gene expression changes (e.g., less chromatin accessibility associated with decreased gene expression due to increased repressive mark H3K27me3 by UTX deficiency). As illustrated in Fig. 9A, the decreases in read densities of genes of two selective clusters based on variable degree of peaks in UTXKO iBAT were highly associated with the down-regulations of the corresponding gene expression, including *Prdm16* (shown in red box) and several other BAT-specific genes (e.g., *Pgc1α*, *Pgc1β*, *Pparα* etc.). Indeed, our ATAC-seq data demonstrated that deleting *Utx* in brown fat resulted in a more closed chromatin structure at both *Prdm16* (Fig. 9B) and *Pgc1α* (Supplementary Fig. 23A) loci, as well as other BAT-specific genes such as *Ucp1* and *Pgc1β* (Supplementary Fig. 23B–C), which could contribute to the suppressed expression of these genes in iBAT of UTXKO mice (Fig. 1F). Our data indicate that both *Prdm16* and *Pgc1α* could be downstream targets of UTX via modulating their chromatin structure. Since PRDM16 regulates BAT-muscle switch in BAT[44], we thus further studied whether PRDM16 could mediate the BAT-to-myocyte remodeling observed in HFD-fed, UTXKO and D1KO mice.

*Prdm16* and *Myod1* are mutually exclusive in lineage determination of brown adipocytes vs. myocytes[57]. We found that *Prdm16* expression was downregulated in iBAT of HFD-fed mice (Fig. 9C), This may be due to a decreased binding of UTX to *Prdm16* promoter as shown in ChIP assays (Fig. 9D). Conversely, the level of H3K27me3, which was regulated by UTX[14], was increased at *Prdm16* promoter (Fig. 9E). To confirm whether *Prdm16* expression is indeed regulated by UTX through

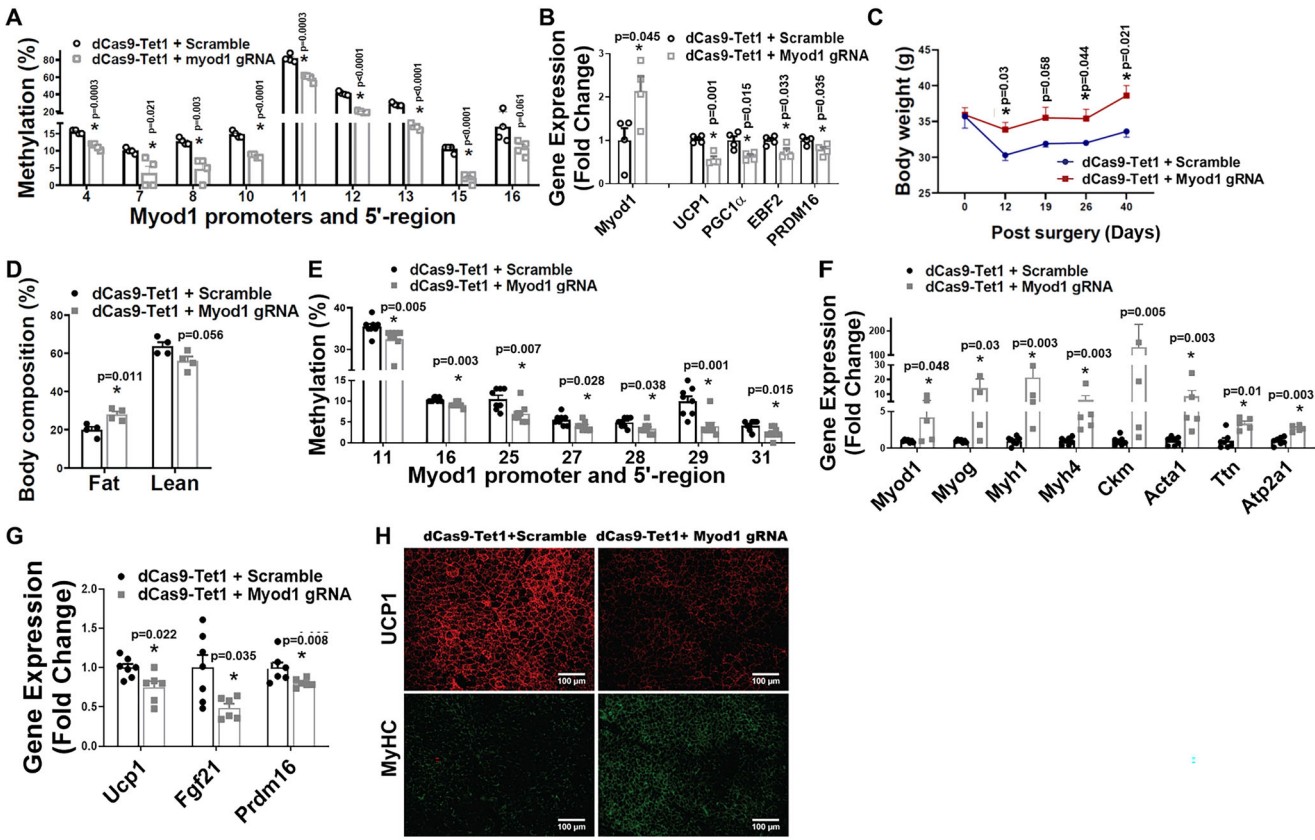

**Fig. 8 Specifically reducing DNA methylation at *Myod1* promoter in iBAT of mice induces BAT-to-myocyte switch. A** Pyrosequencing analysis of DNA methylation at *Myod1* promoter in BAT1 adipocytes transfected with lentiviral vectors expressing dCas9-TET1CD along with lentiviral vectors expressing either *Myod1*-targeting gRNA or scramble non-targeting gRNA ($n = 4$/group). *indicates statistical significance between the two groups by two-tailed unpaired Student's *t*-test. **B** Quantitative PCR analysis of Myod1 and BAT-specific gene expression in BAT1 adipocytes transfected with lentiviral vectors expressing dCas9-TET1CD along with lentiviral vectors expressing either *Myod1*-targeting gRNA or scramble non-targeting gRNA ($n = 4$/group). *indicates statistical significance between the two groups by two-tailed unpaired Student's *t*-test. For (**A**, **B**), 4-day differentiated BAT brown adipocytes were transfected with lentiviral vectors FUW-dCas9-TET1CD along with lentiviral vectors pgRNA-mCherry encoding either scramble-gRNA or Myod1-targeting gRNA using Amaxa Nucleofector II Electroporator with an Amaxa cell line nucleofector kit L. Cells were harvested 2 days after for pyrosequencing (**A**) or gene expression (**B**) analysis. **C**, **D** Body weight (**C**) and Body composition (**D**) in mice with iBAT injection of lentiviruses expressing dCas9-TET1CD plus lentiviruses expressing either targeting Myod1-gRNA-mCherry or non-targeting scramble-gRNA-mCherry ($n = 4$/group). *indicates statistical significance between the two groups by two-tailed unpaired Student's *t*-test. **E**–**G** Methylation levels at *Myod1* promoter (**E**, $n = 8$/group), Myogenic marker gene expression (F, $n = 7$ for dCas9 + scramble, and 5 for dCas9 + Myod1 gRNA), and BAT-specific gene expression (**G**, $n = 7$ for dCas9+scramble, and 6 for dCas9 + Myod1 gRNA) in iBAT of mice with iBAT injection of lentiviruses expressing dCas9-TET1CD plus lentiviruses expressing either targeting Myod1-gRNA-mCherry or non-targeting scramble-gRNA-mCherry. *Indicates statistical significance between the two groups by Mann–Whitney's nonparametric *U* test in (**E**), (**F**) and (**G**). (**H**) Representative IHC staining of UCP1 (upper panel) and MyHC (lower panel) in iBAT of mice with iBAT injection of lentiviruses expressing dCas9-TET1CD plus lentiviruses expressing either targeting Myod1-gRNA-mCherry or non-targeting scramble-gRNA-mCherry ($n = 3$ replicates). For (**C**–**H**), 3-month-old chow-fed male C57BL/6J mice were bilaterally injected with lentiviruses expressing dCas9-TET1CD plus lentiviruses expressing either targeting Myod1-gRNA-mCherry or non-targeting scramble-gRNA-mCherry into iBAT for up to 2 months. All data are expressed as mean ± SEM.

modulating of H3K27me3 levels, we knocked down *Utx* in BAT1 brown adipocytes and measured H3K27me3 at *Prdm16* promoter via ChIP assay following stimulation by the β-adrenergic agonist isoproterenol. We found that H3K27me3 level was increased at *Prdm16* promoter in isoproterenol-treated BAT1 brown adipocytes with *Utx* knockdown (Fig. 9F). Thus, our data suggest that UTX promotes *Prdm16* expression via demethylating H3K27me3 at *Prdm16* promoter.

We further determined whether UTX-PRDM16 axis regulates *Myod1* promoter methylation. Indeed, knockdown of *Utx* in BAT1 brown adipocytes resulted in reduced DNA methylation at a number of CpG sites at *Myod1* promoter (Fig. 9G), which was associated with increased *Myod1* expression (Fig. 9H). Similarly, BAT1 brown adipocytes with *Prdm16* knockdown also exhibited decreased DNA methylation at *Myod1* promoter (Fig. 9I), which

was associated with increased *Myod1* expression (Fig. 9J); while overexpression of *Prdm16* significantly increased DNA methylation at Myod1 promoter (Fig. 9K).

To determine how PRDM16 regulates DNA methylation at *Myod1* promoter, we knocked down *Prdm16* and measured DNMT1 binding to *Myod1* promoter via ChIP assays in BAT1 brown adipocytes with or without isoproterenol stimulation. We found that stimulation of BAT1 brown adipocytes with the β-adrenergic agonist isoproterenol significantly induced DNMT1 binding to *Myod1* promoter; whereas *Prdm16* knockdown in BAT1 brown adipocytes suppressed both basal- and isoproterenol-induced DNMT1 binding to *Myod1* promoter (Fig. 10A), suggesting that PRDM16 is required for the full ability of DNMT1 in maintaining DNA methylation at *Myod1* promoter in brown adipocytes.

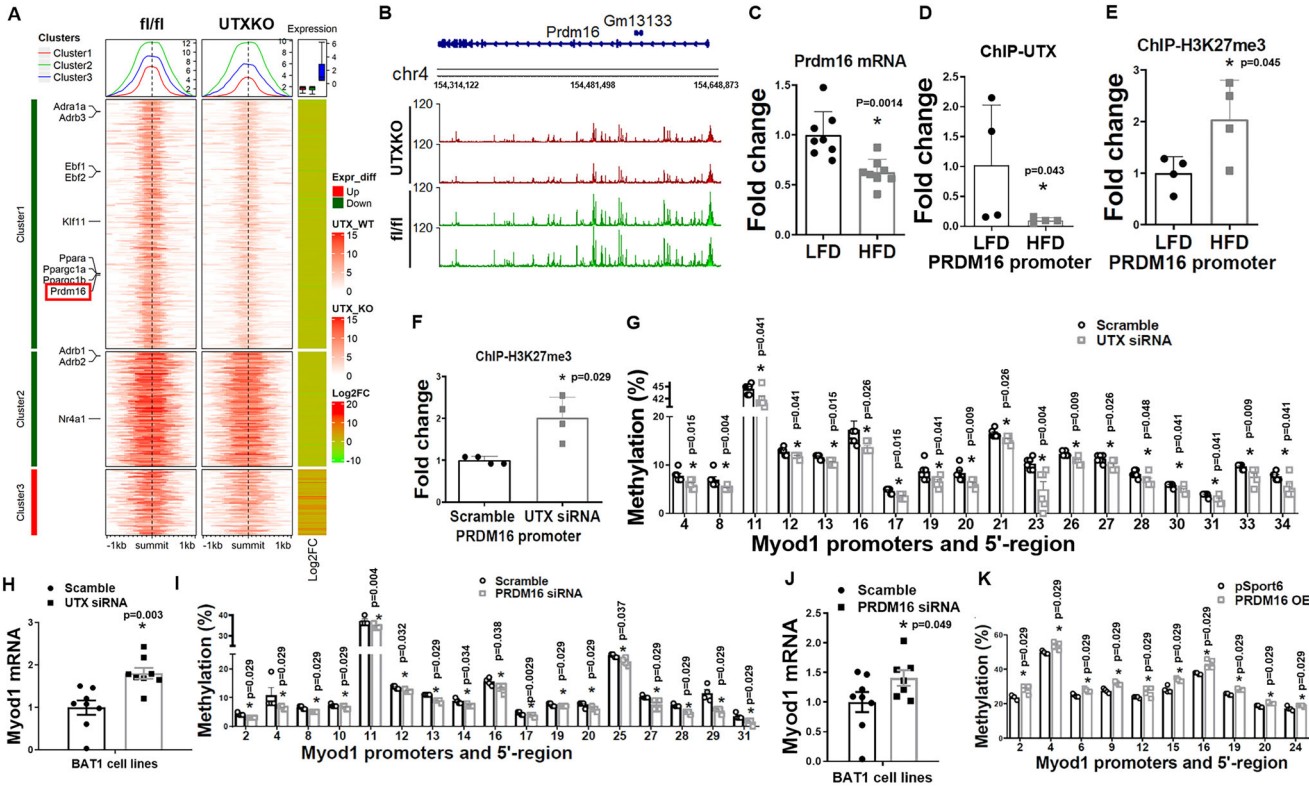

**Fig. 9 DNMT1 silences *Myod1* expression via interacting with PRDM16. A** Comparison of genome-wide alterations in chromatin accessibility landscape assessed by ATAC-seq with the corresponding gene expression assessed by RNA-seq in iBAT of UTXKO and fl/Y mice fed HFD for 12 weeks (*n* = 3 replicates per group). **B** ATAC-seq analysis of chromatin accessibility at Prdm16 gene locus in iBAT of UTXKO and fl/Y mice fed HFD for 12 weeks (*n* = 3 replicates per group). **C** Quantitative RT-PCR analysis of *Prdm16* mRNA in iBAT of LFD- or HFD-fed mice (*n* = 8/Group). *Indicates statistical significance between the two groups by two-tailed unpaired Student's *t*-test. **D, E** ChIP assay of UTX binding to *Prdm16* promoter (**D**, *n* = 4/group) and ChIP assay of H3K27me3 levels at *Prdm16* promoter (**E**, *n* = 4/group) in iBAT of LFD- or HFD-fed mice. *Indicates statistical significance between the two groups by Mann–Whitney's nonparametric *U* test in (**D**) and two-tailed unpaired Student's *t*-test in (**E**). **F** ChIP assay of H3K27me3 levels at *Prdm16* promoter in control or *Utx* knockdown BAT1 brown adipocytes treated with isoproterenol (*n* = 4/Group). *Indicates statistical significance between the two groups by Mann–Whitney's nonparametric *U* test. **G–H** Pyrosequencing analysis of DNA methylation at *Myod1* promoter (**G**, *n* = 6/group) and *Myod1* expression (**H**, *n* = 8/group) in BAT1 brown adipocytes transfected with scramble or *Utx* siRNA. *Indicates statistical significance between the two groups by Mann–Whitney's nonparametric *U* test in (**G**) and two-tailed unpaired Student's *t*-test in (**H**). **I, J** Pyrosequencing analysis of DNA methylation at *Myod1* promoter (**I**, *n* = 4/group) and Myod1 expression (**J**, *n* = 8 for Scramble and 7 for Prdm16 siRNA) in BAT1 brown adipocytes transfected with scramble or *Prdm16* siRNA. *Indicates statistical significance between the two groups by Mann–Whitney's nonparametric *U* test in (**I**) and two-tailed unpaired Student's *t*-test in (**J**). **K** Pyrosequencing analysis of DNA methylation at *Myod1* promoter in BAT1 brown adipocytes transfected with pSPORT6 or pSPORT6 encoding *Prdm16* overexpressing plasmids (*n* = 4/group). *indicates statistical significance between the two groups by Mann–Whitney's nonparametric *U* test. All data are expressed as mean ± SEM.

To further determine whether PRDM16 directly interacts with DNMT1, we overexpressed Flag-PRDM16 and DNMT1 in HEK293T cells and conducted co-immunoprecipitation assay. Indeed, immunoprecipitation of PRDM16 with a Flag antibody pulled down DNMT1 protein (Fig. 10B). To further determine the domains that dictate the interaction between PRDM16 and DNMT1, we expressed various truncated fragments of PRDM16 and DNMT1 to perform co-immunoprecipitation assays in HEK293T cells. PRDM16 is a transcriptional cofactor that contains six major domains, including a PR/SET domain (PR) with monomethyltransferase activity and two zinc-finger domains (ZF1/ZF2) known to interact with PGC1α[58–60]. Other domains include a proline-rich domain (PRR), a repressor domain (RD), and a C-terminal acidic domain (AD), although the function of these domains has not been fully elucidated[58–60]. Interestingly, immunoprecipitation of DNMT1 followed by immunoblotting of HA-PRDM16 fragments with a HA antibody revealed that DNMT1 protein interacted with both PR and ZF2 domains of PRDM16 (Fig. 10C).

Meanwhile, DNMT1 also has six functional domains including the N-terminal independently folded domain (NTD) that is known to interact with various protein that regulates DNMT1 functions, replication foci-targeting sequence (RFTS) domain, a Zn-finger like CXXC motif, two bromo adjacent homology (BAH1 and BAH2) domains, and the catalytic domain[61,62]. Using a similar approach, we performed immunoprecipitation of HA-DNMT1 fragments with HA antibodies followed by immuno-blotting with PRDM16 antibodies, and found that PRDM16 interacted with the NTD domain of DNMT1 protein (Fig. 10D).

Recent data suggest that the NTD domain on DNMT1 regulates its protein stability by interacting with the protein lysine methyltransferase SET7 and the serine/threonine kinase AKT1[63]. To study whether the observed interaction between PRDM16 and the NTD domain on DNMT1 regulates DNMT1 protein stability, we co-transfected PRDM16 and DNMT1 into HEK293T cells and treated cells with the protein translation inhibitor cycloheximide. As shown in Fig. 10E, co-transfection of PRDM16 significantly prevented DNMT1 protein degradation

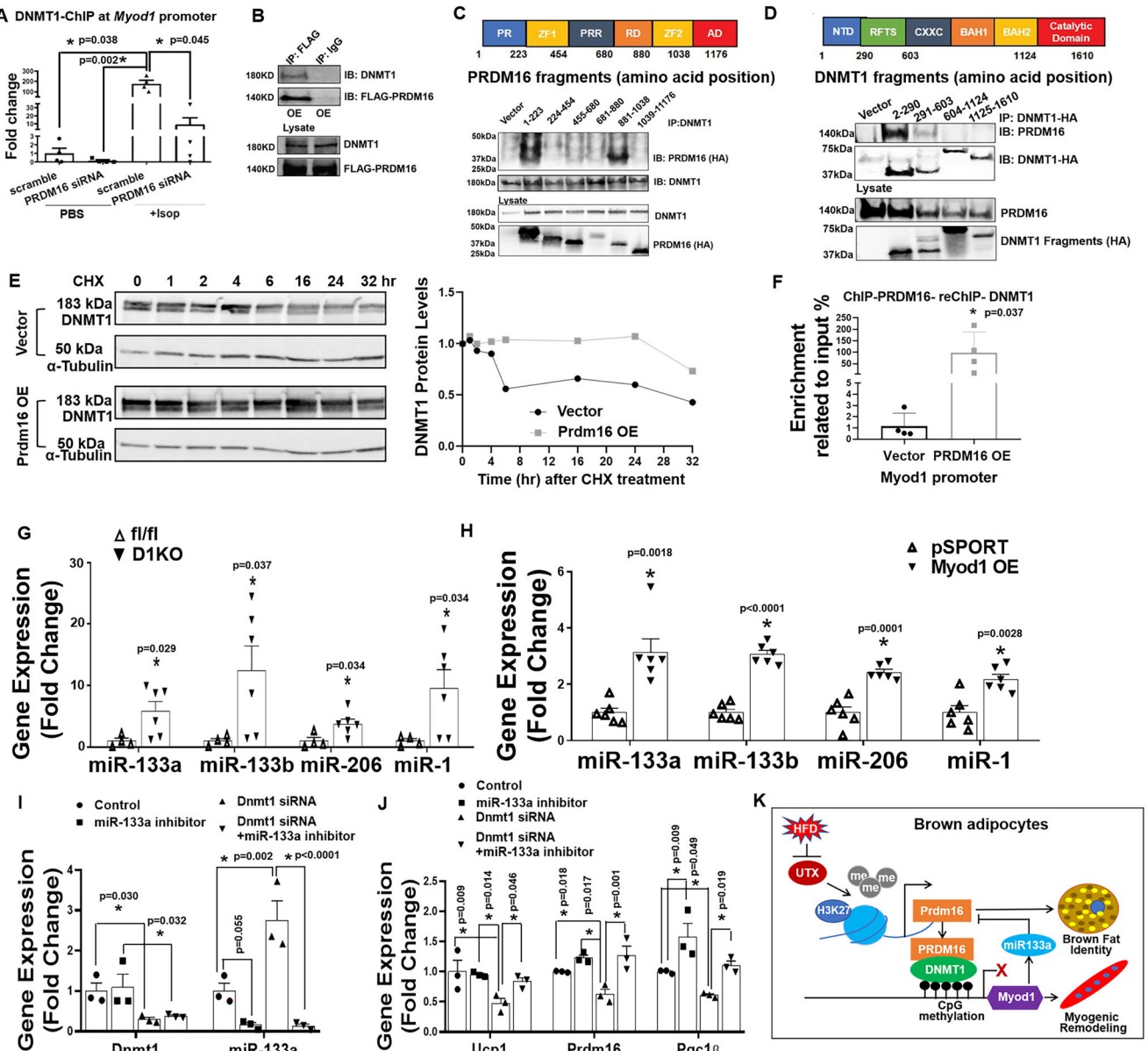

following cycloheximide treatment, indicating that the presence of PRDM16 increases DNMT1 protein stability, possibly through binding to the NTD domain on DNMT1. In addition, ChIP and Re-ChIP assay on *Myod1* promoter with serial immunoprecipitation against PRDM16 and DNMT1 suggest that the presence of PRDM16 significantly increased both PRDM16 and DNMT1 binding on *Myod1* promoter (Fig. 10F), which could in turn regulate *Myod1* promoter DNA methylation and *Myod1* expression. Thus, our data suggest that the increased binding of PRDM16 with DNMT1 at *Myod1* promoter when PRDM16 is present is probably due to two mechanisms. First, PRDM16 could increase DNMT1 protein stability via binding to the NTS domain on DNMT1, as shown in Fig. 10E; and second, PRDM16 could increase the recruitment of DNMT1 to the Myod1 promoter, as shown in the ChIP and Re-ChIP assay in Fig. 10F.

Our data so far suggest that the BAT-to-myocyte remodeling in *Utx*-deficient brown adipocytes is probably mediated via reduced chromatin accessibility at *Prdm16* locus, which leads to reduced *Prdm16* expression. This leads to upregulated myogenic gene expression through reduced interaction and binding between PRDM16 and DNMT1 at *Myod1* promoter. In addition,

reduced *Prdm16* expression in *Utx*-deficient brown adipocytes also downregulates thermogenic program in brown fat.

Similarly, the deletion of *Dnmt1* in *Dnmt1*-deficient brown adipocytes could result in upregulation of myogenic gene expression via reduced *Myod1* promoter DNA methylation. However, how upregulated *Myod1* expression in *Dnmt1*-deficient brown adipocytes results in reduced thermogenic gene expression is less clear. Recent data suggest that *Myod1* represses *Prdm16* in muscle progenitor cells via interaction with the E2F transcription factor 4 (E2F4)/RB transcriptional corepressor like 1 (RBL1/ p107)/ RB transcriptional corepressor like 2 (RBL2/p130) transcription repressor complex at *Prdm16* promoter[64]. Thus, we further explored whether MYOD1 could repress *Prdm16* via this potential pathway in BAT1 cells. We found that over-expressing *Myod1* in BAT1 cell led to increased binding of E2F4, RBL1 and RBL2 at *Prdm16* promoter (Supplementary Fig. 24A–C). However, knocking down of *E2f4*, *Rbl1* or *Rbl2* in BAT1 cells (Supplementary Fig. 24D–F) did not significantly change basal expression of BAT-specific genes, nor did they reverse *Myod1* overexpressoin-induced suppression of BAT-specific gene expression when *E2f4*, *Rbl1* or *Rbl2* were knocked

**Fig. 10 PRDM16 interacts with DNMT1 to maintain brown adipocyte function. A** ChIP assay of DNMT1 binding to *Myod1* promoter in control or *Prdm16* knockdown BAT1 brown adipocytes treated with or without isoproterenol (*n* = 4/group). Data are expressed as mean ± SEM. Indicates statistical significance between different treatments analyzed by Kruskal–Wallis non-parametric ANOVA H test by rank followed by Pairwise Comparisons test between groups, H(3) = 9.816, *p* = 0.020. **B** Co-IP of DNMT1 and FLAG-PRDM16 in HEK293T cells. Data are representative from two independent experiments. **C** Co-IP of DNMT1 and various fragments of PRDM16. HA-tagged fragments of PRDM16 were expressed along with full-length DNMT1 in HEK293T cells. Cell lysates were immunoprecipitated with anti-DNMT1 antibodies followed by immunoblotting with HA or DNMT1 antibodies. Color-coded domain architecture of PRDM16 shows a PR/SET domain (PR), an N-terminal zinc-finger domain containing seven C2H2 zinc finger motifs (ZF1), a proline rich domain (PRR), a repression domain (RD), a second C-terminal zinc-finger domain containing three C2H2 zinc finger motifs (ZF2), and an acidic activation domain (AD). Data are representative from two independent experiments. **D** Co-IP of PRDM16 and various fragments of DNMT1. HA-tagged fragments of DNMT1 were expressed along with full-length PRDM16 in HEK293T cells. Cell lysates were immunoprecipitated with anti-HA antibodies followed by immunoblotting with HA or PRDM16 antibodies. Color-coded domain architecture of DNMT1 shows the N-terminal independently folded domain (NTD), replication foci-targeting sequence (RFTS) domain, a Zn-finger like CXXC motif, two bromo adjacent homology (BAH1 and BAH2) domains, and the catalytic domain. Data are representative from two independent experiments. **E** DNMT1 protein levels in Prdm16-overexpressed HEK293T cells treated with cycloheximide (CHX) (60 µg/ml) for various time. Data are representative from two independent experiments. **F** The interaction between PRDM16 and DNMT1 on *Myod1* promoter in *Prdm16* overexpressed BAT1 brown adipocytes measured by ChIP and Re-ChIP assay via sequential immunoprecipitation of PRDM16 and then DNMT1 (*n* = 4/group). Data are expressed as mean ± SEM. *Indicates statistical significance between two groups by Mann–Whitney's nonparametric *U* test. **G, H** Expression of *miR-133a*, *miR133b*, *miR-206* and *miR-1* in iBAT of female D1KO and fl/fl mice fed with a regular chow diet (**G**, *n* = 4 for fl/fl and 6 for D1KO) and in BAT1 brown adipocytes with *Myod1* overexpression (**H**, *n* = 6/group). Data are expressed as mean ± SEM. *indicates statistical significance between the two groups by Mann–Whitney's nonparametric *U* test in (**G**) and by two-tailed unpaired student's *t*-test in (**H**). **I, J** *Dnmt1* and *miR-133a* expression (**I**) and BAT-specific gene expression (**J**) in BAT1 brown adipocytes transfected with *Dnmt1* siRNA, *miR-133a* inhibitor or both (**J**) (*n* = 3/group). Data are expressed as mean ± SEM. *Indicates statistical significance analyzed by one-way ANOVA followed by Fisher's LSD multiple comparisons test. In (**I**), for *Dnm1* expression, F = (3,8) = 4.62, *p* = 0.037; for *miR-133* expression, F(3,8) = 21.370, *p* < 0.0001. In (**J**), for *Ucp1* expression, F = (3,8) = 4.827, *p* = 0.033; for *Prdm16* expression, F = (3,8) = 10.863, *p* = 0.003; for *Pgc1β* expression, F(3,8) = 11.213, *p* = 0.003. **K** Schematic illustration of the interaction between UTX-regulated PRDM16 and DNMT1 in the maintenance of brown fat identity and suppression of myogenic remodeling in mature brown adipocytes. In brief, in mature brown adipocytes, UTX maintains the persistent demethylation of the repressive mark H3K27me3 at *Prdm16* promoter, leading to high expression of *Prdm16*; PRDM16 then recruits the DNA methyltransferase DNMT1 to *Myod1* promoter, causing *Myod1* promoter hypermethylation, and suppressing *Myod1* expression. In addition, reduced *Myod1* expression relieves the inhibition on *Prdm16* by miR-133, further increasing *Prdm16* expression. The interaction between PRDM16 and DNMT1 coordinately serves to maintain brown adipocyte identity while repressing myogenic remodeling in mature brown adipocytes, thus promoting their active brown adipocyte thermogenic function. Suppressing this interaction by HFD feeding induces brown adipocyte-to-myocyte remodeling, which limits brown adipocyte thermogenic capacity and compromises diet-induced thermogenesis, leading to the development of obesity.

down individually or in combination (Supplementary Fig. 24G–J), These data suggest that unlike its role in muscle progenitor cells, the E2F4/RBL1/RBL2 repressor complex does not play an important role in mediating MYOD1's suppression of BAT-specific gene expression in BAT1 brown adipocytes.

Alternatively, microRNA-1 (miR-1), miR-206, miR-133a, and miR-133b are specifically expressed in skeletal muscle that are important in regulating muscle development[65–68], and their expression is regulated by MYOD1[66,67]. Recently, it has been reported that miR-133a/133b negatively regulates BAT-specific gene expression in muscle satellite cells by specifically targeting *Prdm16*[68]. Interestingly, we found that the expression of these microRNAs was significantly increased in iBAT of D1KO mice (Fig. 10G) and in BAT1 cells overexpressing *Myod1* (Fig. 10H). In addition, *Dnmt1* knocking down in BAT1 cells also significantly upregulated *miR-133a* expression (Fig. 10I); whereas miR-133a inhibition in BAT1 cells upregulated BAT-specific gene expression (Fig. 10J). Importantly, knocking down *Dnmt1* in BAT1 cells significantly suppressed BAT-specific gene expression, including *Ucp1*, *Prdm16,* and *Pgc1β*, which was reversed by *miR-133a* knocking down (Fig. 10I, J). Thus, our data suggest that increased *Myod1* expression in *Dnmt1*-deficient brown adipocytes results in upregulation of *miR-133*, which in turn suppresses *Prdm16* expression, leading to downregulation of thermogenic gene expression.

Changes in DNA methylation have been shown to modulate histone methylation, which may act cooperatively to influence chromatin structure and thereby regulate gene expression[69]. We, therefore, interrogated the effects of DNMT1-mediated DNA methylation on histone modifications at the *Myod1* promoter by surveying the status of several common histone marks using ChIP assays. We found that *Dnmt1* knockdown markedly reduced histone repressive mark histone 3 lysine 9 trimethylation (H3K9me3) while simultaneously enhancing transcriptional active mark H3K4 acetylation (H3K9ac) (Supplementary Fig. 25A–B), without affecting other histone marks such as H3K9ac, H3K4me3, and H3K27me3 (Supplementary Fig. 25C–E). The reciprocal changes of decreased H3K9me3 and increased H3K4ac could contribute to increased chromatin accessibility at the *Myod1* gene locus in iBAT of the UTXKO mice as revealed by ATAC-seq analysis (Supplementary Fig. 25F), which may account for the upregulation of *Myod1* expression that mediates BAT-to-myocyte remodeling in *Utx*- and *Dnmt1*-deficient BAT.

## Discussion

We have previously shown that the histone demethylase UTX promotes thermogenic program in brown adipocytes in vitro[13]; here we have determined the role of *Utx* in the regulation of brown fat thermogenesis and energy metabolism in vivo using genetic models and interrogated the underlying mechanism. We found that mice with *Utx* deficiency in brown adipocytes had impaired BAT thermogenesis and were susceptible to diet-induced obesity. Interestingly, we discovered that *Utx*-deficient brown adipocytes not only displayed suppressed brown adipocyte thermogenic features, they also underwent a BAT-to-myocyte-like remodeling, characterized by a profound upregulation of myogenic markers. Interestingly, the BAT-to-myocyte remodeling was only observed in brown adipocytes from UTXKO mice, where *Utx* deletion occurred in mature brown adipocytes with the use of *Ucp1-Cre*, but not in brown adipocytes from MUTXKO mice, where *Utx* was deleted in *Myf5*-expressing myoblast precursor cells with the use of *Myf5-Cre*. Thus, our data suggest that UTX may be dispensable in the BAT-skeletal muscle lineage determination during BAT development, but it is necessary in

maintaining brown adipocyte identity and suppressing myogenic remodeling in mature brown adipocytes.

Intriguingly, our parallel study involving another genetic model D1KO mice with brown adipocyte-specific *Dnmt1* deletion revealed that D1KO mice strikingly mimicked UTXKO mice in their myocyte-like brown fat remodeling, as well as in metabolic phenotypes of their susceptibility to diet-induced obesity, suggesting that both UTX and DNMT1 are involved in regulating BAT-myocyte remodeling in brown adipocytes.

It has been recently demonstrated that adult humans also have functional BAT that is inversely correlated with body weight[7–9,70–73], and repeated cold exposure treatments activate human BAT that is associated with an increase in energy expenditure[74,75]. Thus, BAT may present a promising target for the treatment of obesity and other metabolic diseases in humans. Interestingly, H3K27me3, the target of UTX, is an inheritable epigenetic pattern that is associated with obesity and type 2 diabetes[76]. In addition, Kabuki syndrome is a rare genetic disease with obesity and type 2 diabetes as frequently reported symptoms among others, and *Utx* is reported as one of the genes to be mutated[77–82]. On the other hand, the association of *Dnmt1* polymorphism with obesity has also been reported in human population[83]. Thus, it is possible that the UTX/DNMT1 pathway may also regulate obesity and associated metabolic diseases in humans.

Brown fat and skeletal muscle share the same developmental origin and are derived from *myf5*-expressing myoblast precursors[21,44,84]. It is well documented that PRDM16 and MYOD1 function as reciprocal master regulators to control the cell lineage determination between brown adipogenesis and myogenesis. PRDM16 promotes BAT-specific gene expression by forming transcriptional complexes with CCAAT/enhancer binding protein β (C/EBPβ), PPARγ and PGC1α in the initiation of BAT development and subsequent maintenance of its identify;[58] whereas MYOD1 acts downstream of MYF5 to initiate myogenic differentiation[85,86].

We found that BAT from UTXKO mice displayed downregulated expression of *Prdm16*, and *Utx* knockdown in BAT1 brown adipocytes resulted in enriched H3K27me3 at *Prdm16* promoter. Indeed, ATAC-seq analysis revealed that chromatin of Prdm16 was less accessible in iBAT of UTXKO mice than that of control mice. These data suggest that *Prdm16* is a downstream target of UTX by epigenetic modulation of H3K27me3 at *Prdm16* promoter. Downstream of PRDM16, it has been shown that PRDM16 interacts with various proteins to form transcriptional complexes that mediates its effects. The ZF1/ZF2 domains of PRDM16 have been shown to interact with PPARγ[44], PGC1α[59], C/EBPβ[87], the histone H3 lysine 9 (H3K9) methyltransferase euchromatic histone methyltransferase 1 (EHMT1)[88], and mediator complex subunit 1 (MED1)[89,90] to mediate PRDM16's effect on maintaining brown fat identity. In addition, the PLDLS motif within the RD domain of PRDM16 has been shown to interact with C-terminal-binding protein-1 (CtBP-1) and CtBP-2 to repress WAT-specific gene expression[59]. However, the function of the N-terminal PR/SET domain in PRDM16 is not well defined.

On the other hand, recent data suggest that the NTD domain on DNMT1 regulates its protein stability by interacting with the protein lysine methyltransferase SET7 and the serine/threonine kinase AKT1[63]. Here we found that PRDM16 utilized its PR and ZF2 domains to directly interact with DNMT1 on its NTD domain, which, through increasing DNMT1 protein stability and recruiting DNMT1 to *Myod1* promoter, resulting in *Myod1* promoter DNA hypermethylation, thus suppressing its expression, underscoring the importance of the PR/SET and ZF2 domains on PRDM16, and NTD domain on DNMT1 in

regulating *Myod1* promoter DNA hypermethylation and *Myod1* expression. This was further confirmed by CRISPR/gRNA-guided approach to specifically induce demethylation at *Myod1* promoter in brown adipocytes, which resulted in increased *Myod1* expression, leading to a BAT-to-myocyte remodeling process.

Our study also demonstrates that this epigenetic event that controls BAT-myocyte remodeling may also take place in the context of diet-induced obesity and regulates diet-induced thermogenesis and systemic energy metabolism. For instance, we found that HFD decreased the binding of UTX to *Prdm16* promoter region, thus increasing H3K27me3 levels at *Prdm16* promoter. This resulted in downregulation of *Prdm16* gene expression. Reduced PRDM16 presence would decrease the recruitment of DNMT1 to *Myod1* promoter, leading to increased *Myod1* expression and initiation of BAT-to-myocyte remodeling in BAT of HFD-fed mice. These myocyte-like brown adipocytes appeared to exhibit decreased energy expenditure as evidenced by decreased OCR in *Utx*- or *Dnmt1*-deficient as well as HFD-fed brown adipocytes. It has been well documented that long-term HFD feeding causes WAT remodeling, as evidenced by enlarged adipocytes, exaggerated inflammation, inappropriate fibrosis, and impaired angiogenesis[91,92], leading to WAT dysfunction. However, much less is known about the impact of long-term HFD feeding on brown adipocytes. Here we show that long-term HFD causes brown fat remodeling into a myocyte-like characteristic. The induction of these myocyte-like brown adipocytes may reduce the thermogenic capacity of brown fat, leading to reduced energy expenditure and obesity.

How HFD induces such epigenetic alterations that lead to BAT-to-myocyte remodeling in BAT is not clear. We found that UTX protein level was downregulated by long-term HFD feeding. Interestingly, UTX was predicted to be ubiquitinated at lysine 225[93]. Further studies are required to investigate whether UTX ubiquitination occurs in iBAT of HFD-fed mice that may contribute its downregulation. On the other hand, UTX catalyzes the removal of the repressive chromatin mark H3K27me3[14], and H3K27 methylation has been linked to the control of cellular differentiation[94,95]. In addition, cellular oxygen level also determines cell fate, and hypoxia is important in maintaining cells at an undifferentiated, precursor-like or stem cell-like state[96]. Interestingly, recent data suggest that UTX, but not its paralog KDM1 lysine (K)-specific demethylase 6B/Jumonji domain-containing 3 (KDM6B/JMJD3) serves as s cellular oxygen sensor; hypoxia and loss of UTX similarly cause persistent increase in H3K27 methylation in key regulators of cellular differentiation, which in turn blocks cellular differentiation[97]. Hypoxia develops in obese white adipose tissue with the expansion of tissue mass, and contributes to adipose tissue dysfunction, including inflammation, fibrosis, and insulin resistance[98]. Recent data also suggest that obesity induces capillary rarefaction and functional hypoxia in BAT, leading to a BAT "whitening" phenotype[99]. Thus, it is possible that hypoxic conditions induced in obese BAT may inhibit UTX activity, causing a persistent increase in H3K27 methylation at promoters of key regulators of differentiation. This in turn reverses these brown adipocytes to a less-differentiated, precursor-like phenotype, leading to a loss of brown adipocyte thermogenic feature and simultaneous upregulation of myogenic gene expression in *Utx*-deficient brown adipocytes, as observed in the current study.

Along the course of our study, a recent report brought us attention in which deletion of *Utx* in all adipocytes including both brown and white adipocytes prevents HFD-induced obesity[100]. The exact reason for the discrepancy between this paper's observation and ours is not clear, but may lie within the different models being tested and the realm of adipocytes (for example, brown or white adipocytes) that had *Utx* deletion. For one,

Ota et al generated genetic model of *Utx* deficiency in all adipocytes by crossing *Utx*-floxed mice with *aP2-Cre* mice. It is noteworthy that employing *aP2-Cre* line may also result in deletion of *Utx* in macrophages due to the presence of aP2 in macrophages[101]. It is possible that *Utx* deficiency in macrophages may alter inflammatory status that in turn affects adipocyte metabolism through a paracrine action; as infiltration of macrophage into adipose tissue during the development of obesity has been well documented[102]. Indeed, specific deletion of *Utx* in macrophages with the use of lysozyme-Cre has been shown to increase M2 macrophage polarization, inhibit white adipocyte differentiation and promote brown adipocyte thermogenesis, leading to decreased adiposity in mice[103]. Thus, the obesity-resistant phenotype observed by Ota et al.[100] may be due to the *aP2-Cre* used that resulted in *Utx* deletion also in macrophages. In addition, we also found that whereas mice with mature brown adipocyte-specific deletion of *Utx* using the *Ucp1-Cre* exhibited a BAT-to-myocyte remodeling and were prone to diet-induced obesity, mice with *Utx* deletion in *Myf5*+-cells did not show any phenotypic differences when fed HFD, despite the fact that brown adipocytes derive from *Myf5*+ lineage precursors[44]. Thus, the differences in the Cre line used may contribute to the differential phenotypes observed between our animal models and those reported by Ota et al.[100].

In the current study, we found that both male and female UTXKO mice showed an obesity-prone phenotype when fed HFD, albeit with much milder phenotype observed in female UTXKO mice. In contrast, male and female D1KO mice exhibited relatively similar metabolic phenotypes including diet-induced obesity and insulin resistance. Thus, our data suggest sexual dimorphism is more evident in UTXKO mice than in D1KO mice. The exact reason is not clear. Sexual dimorphism frequently occurs in metabolic phenotypes of both humans and rodents. For one, males and females have different fat composition and distribution in humans[104,105]. This might be due to differential lipid metabolism between the two genders[106]. Another potential mechanism may be attributed to the sex hormone estrogen and its receptors that have been shown to play a pivotal role in various metabolic pathways[107]. Our data in the current study suggest that UTX may potentially be acting as an upstream regulator of DNMT1. Thus, UTX signaling may be more susceptible to modification by other genetic or environmental cues than the downstream DNMT1. It is possible these factors may regulate UTX down-stream signaling other than DNMT1 in UTXKO mice that may contribute to the phenotype difference observed in UTXKO and D1KO mice.

In summary, we have identified a BAT-myocyte remodeling process that appears to be present in mature brown adipocytes; this process is regulated by the interaction of epigenetic pathways that regulate histone and DNA methylation (Fig. 10K). In brief, in mature brown adipocytes, UTX maintains the persistent demethylation of the repressive mark H3K27me3 at *Prdm16* promoter, leading to high expression of Prdm16. PRDM16 then recruits the DNA methyltransferase DNMT1 to *Myod1* promoter, causing *Myod1* promoter hypermethylation, and suppressing *Myod1* expression in mature brown adipocytes. On the other hand, reduced Myod1 expression in turn down-regulates miR-133a/b, which further leads to increased Prdm16 expression. Thus, the interaction between PRDM16 and DNMT1 coordinately serves to maintain brown adipocyte identity while repressing myogenic remodeling in mature brown adipocytes, thus promoting their active brown adipocyte thermogenic function. The suppression of this pathway by HFD feeding results in the induction of BAT-to-myocyte remodeling, which could limit the thermogenic capacity of brown fat and thereby compromise diet-induced thermogenesis, leading to the development of obesity (Fig. 10K).

## Methods

**Mice.** Mice with brown adipocyte-specific *Utx* knockout were generated by crossing *Utx*-floxed mice (Jackson Laboratory, stock No. 021926)[15] with *Ucp1-Cre* mice (Jackson Laboratory, Stock No. 024670)[16], where *Ucp1* is specifically expressed in brown adipocytes and UCP1-positive beige adipocytes (UTXKO, *Ucp1^Cre::Utx^fl/y* for male and *Ucp1^Cre::Utx^fl/fl* for female). The littermates *Utx^fl/Y* (fl/Y) or *Utx^fl/fl* (fl/fl) were used as male or female control mice, respectively.

Mice with brown adipocyte-specific DNA methyltransferase 1 (*Dnmt1*) knockout were generated by crossing *Dnmt1*-floxed mice (Mutant Mouse Regional Resource Centers (MMRRC, No. 014114)[41] with *Ucp1-Cre* mice[16] (D1KO, *Ucp1^Cre::Dnmt1^fl/fl*), with littermates *Dnmt1^fl/fl* (fl/fl) as control mice.

Mice with brown adipocyte-specific *Dnmt3a* knockout were generated by crossing *Dnmt3a*-floxed mice (MMRRC No. 029885)[45] with *Ucp1-Cre* mice[16](D3aKO, *Ucp1^Cre::Dnmt3a^fl/fl*), with littermates *Dnmt3a^fl/fl* (fl/fl) as control mice.

Mice with *Utx* deletion in *Myf5*-expressing cells were generated by crossing *Utx*-floxed mice with *Myf5-Cre* mice[22] (Jackson Laboratory, Stock No. 007893) (MUTXKO, *Myf5^Cre::Utx^fl/y* for male and *Myf5^Cre::Utx^fl/fl* for female), with littermates *Utx^fl/Y* (fl/Y) or *Utx^fl/fl* (fl/fl) as male or female control mice, respectively.

D1KO mice with brown adipocytes-specific GFP labeling were generated by triple-crossing *Dnmt1*-floxed mice, *Rosa-Gfp* mice (Jackson Laboratory, Stock No. 006148)[42] and *Ucp1-Cre* mice (D1KO-GFP, *Ucp1^Cre::Dnmt1^fl/fl::Rosa-Gfp^fl/fl*), with littermates *Dnmt1^fl/fl::Rosa-Gfp^fl/fl* as control mice (fl/fl-GFP).

C57BL/6J (B6) and leptin-deficient *ob/ob* mice (Jackson Laboratory, Stock No. 000664 and 000632, respectively) were used in some experiments.

All animal procedures were approved by the Institutional Animal Care and Use Committee at Georgia State University and were in compliance with the Public Health Service and United States Department of Agriculture guidelines.

**Metabolic measurement.** Mice were housed in a temperature- and humidity-controlled animal facility with a 12/12 h light–dark cycle and had free access to water and food (ambient temperature: 20–22 °C, humidity: 30–70%). C57BL//6J wild-type mice, UTXKO, D1KO, and D3aKO mice and their respective littermate controls were weaned onto a regular chow diet (LabDiet 5001, LabDiet, St. Louis, MO, 13.5% calories from fat) or put on a HFD (Research Diets D12492, 60% calorie from fat) diet when they were 5–6 weeks of age. In some of the experiments, a low-fat diet (LFD) (Research Diets D12450B, 10% calorie from fat) was used as a control diet. Various metabolic measurements were conducted as follows. (1) Body weight were measured weekly. (2) Energy expenditure and locomotor activity were measured using PhenoMaster metabolic cage systems (TSE Systems, Chesterfield, MO). Food intake was measured in single-housed animals over seven consecutive days. (3) Body composition representing fat and lean mass was analyzed using a Minispec NMR body composition analyzer (Bruker BioSpin Corporation; Billerica, MA). (4) Fed and fasting glucose was measured by OneTouch Ultra Glucose meter (LifeScan, Milpitas, CA). Glucose and insulin sensitivity was determined by glucose tolerance and insulin tolerance tests (GTT and ITT, respectively) as we previously described[108]. At the end of studies, various tissues including BAT and WAT were collected for further analysis of brown fat/beige adipocyte thermogenic program including gene expression, protein expression and IHC.

**Cold exposure.** UTXKO, D1KO mice and their respective littermate controls underwent a cold challenge (5 °C) for 7 days. At the end of experiment, adipose tissues were dissected for further analysis of brown fat/beige adipocyte thermogenic gene expression, protein expression and IHC. In some cold exposure experiments, a temperature transponder (BioMedic Data Systems, Seaford, DE) was implanted into mouse peritoneal cavity to monitor the body temperature as we previously described[109].

**Quantitative RT-PCR.** Total RNA from tissues or cells was extracted using Tri Reagent kit (Molecular Research Center, Cincinnati, OH)[108]. The mRNA of genes of interest was measured by a one-step quantitative RT-PCR with a TaqMan Universal PCR Master Mix kit (ThermoFisher Scientific, Waltham, MA) using an Applied Biosystems QuantStudio 3 real-time PCR system (ThermoFisher Scientific) as we previously described[108], and was further normalized to the housekeeping gene cyclophilin. For miRNA expression measurement, TaqMan™ MicroRNA Reverse Transcription Kit (ThermoFisher Scientific, #4366596) was used to convert miRNA to cDNA, and miRNA expression was measured with the TaqMan Universal PCR Master Mix kit (ThermoFisher Scientific, Waltham, MA) using Applied Biosystems QuantStudio 3 real-time PCR system (ThermoFisher Scientific), and was further normalized to U6 snRNA. The TaqMan primers/probes for all the genes and miRNAs either purchased from Applied Biosystems (ThermoFisher Scientific) or commercially synthesized were listed in Supplemental Tables 1 and 2.

**Immunoblotting.** Protein levels of gene of interest in adipose tissue were assessed by immunoblotting as we described[12,13,108]. Tissues were homogenized in a modified radioimmunoprecipitation assay (RIPA) lysis buffer supplemented with 1% protease inhibitor mixture and 1% phosphatase inhibitor mixture

(Sigma-Aldrich, St. Louis, MO) and tissue lysates were resolved by SDS-PAGE. Proteins on the gels were transferred to nitrocellulose membranes (Bio-Rad, Hercules, CA), which were then blocked, washed, and incubated with various primary antibodies, followed by Alexa Fluor 680-conjugated secondary antibodies (Life Science Technologies). The blots were developed with a Li-COR Imager System and analyzed with Li-COR Image Studio Software (version 2.1, Li-COR Biosciences, Lincoln, NE). The antibodies were listed in Supplementary Table 3.

**Immunohistochemistry (IHC)**. WAT or BAT tissues were fixed in 10% neutral formalin and embedded in paraffin, which was further cut into 5 μm sections. The sections were either processed for hematoxylin and eosin (H&E) staining or immuno-staining with various antibodies as we previously described[109,110]. The primary antibodies and the secondary antibodies were listed in Supplementary Table 3.

Briefly, to detect myocyte fibers in tissue sections, paraffin-embedded sections were incubated with anti-myosin heavy chain (MyHC) antibodies (MF20, Developmental Studies Hybridoma Bank (DSHB), University of Iowa, Iowa City, IA)[44] overnight at 4 °C and then incubated with anti-mouse secondary antibodies labeled with Alexa fluor 488 (Invitrogen) for 1 h at room temperature and counterstained with 4′,6-diamidino-2-phenylindole (DAPI). For in vitro study, cell cultures were fixed in 10% formalin. Fixed cells were incubated with anti-MyHC antibodies (MF20) overnight at 4 °C and then incubated with anti-mouse secondary antibodies labeled with Alexa fluor 594 (Invitrogen) for 1 h at room temperature and counterstained with DAPI. For IHC, paraffin-embedded sections of iBAT tissues were incubated with anti-UCP1 antibodies (abcam 10983) overnight at 4 °C and then incubated with biotin-conjugated anti-rabbit secondary antibody (Jackson ImmunoResearch, 711-065-152) for 30 min at room temperature. The sections were washed in PBS and incubated with streptavidin-conjugated horseradish Peroxidase (VECTASTAIN® ABC Kit, PK-6100). Then the sections were washed in PBS and incubated with 3, 3-diaminobenzidine (DAB). Histology images were captured using an Olympus DP73 photomicroscope and analyzed by CellSens software (version 1.6) (Olympus, Waltham, MA).

**Cell culture, SiRNA knockdown, overexpression, miRNA inhibition, and oil red O staining**. Immortalized brown fat preadipocytes BAT1 (obtained from Dr. Patrick Seale, University of Pennsylvania)[21,24] were maintained in growth medium (DMEM/F12 containing 10% fetal bovine serum and 1% penicillin/streptomycin) at 37 °C with 5% CO. The cells were differentiated into brown adipocytes as described[13,24]. Briefly, to differentiate brown preadipocytes, 90% confluent cells were cultured in induction medium (growth medium supplemented with 20 nM insulin, 1 nM T3, 125 μM indomethacin, 500 μM isobutylmethylxanthine (IBMX) and 0.5 μM dexamethasone) for two days. After two days, cells were cultured in maintenance medium (growth medium supplemented with 20 nM insulin and 1 nM T3) until experiment.

For siRNA knockdown assays, targeting siRNA and non-targeting scramble control siRNA were purchased from GE Healthcare (mouse *Dnmt1* siRNA-SMARTpool (L-056796-01), *Utx* siRNA-SMARTpool (L-042844-01), *Myod1* siRNA-SMARTpool (L-041113-00), *Prdm16* siRNA-SMARTpool (L-041318-01), *E2f4* siRNA-SMARTpool (L-054294-00), Rbl1 siRNA-SMARTpool (L-042276-00), Rbl2 siRNA-SMARTpool (L-060528-00), and non-Targeting Scramble Control siRNA (D-001810-01)).

For miRNA inhibition experiments, Ambion® Anti-miR™ miRNA Inhibitor Negative Control, Ambion® Anti-miR™ miRNA Inhibitor mmu-miR-133a-5p and Ambion Anti-miR™ miRNA Inhibitor hsa-miR-133a-3p were purchased from ThermoFisher (Assay ID #AM17010, AM11679, and AM10413, respectively).

For plasmids overexpression and sub-cloning, full-length Myod1 overexpressing plasmid was purchased from Addgene (Addgene #8398). Full-length *Prdm16* overexpressing plasmid with an in-frame N-terminal FLAG tag was purchased from Addgene (Addgene #15504). Full-length *Dnmt1* cDNA clone was obtained from Open Biosystems and further subcloned into the pLVX lentiviral expression vector (Clontech) as we previously described[108]. For sub-cloning of *Prdm16* and *Dnmt1* fragments for co-immunoprecipitation experiments, Fragments of *Prdm16* (1–223, 224–454, 455–680, 681–880, 881–1038, and 1039–1176) were PCR-amplified using the full-length Prdm16 plasmids (Addgene #15503) and sub-cloned into XbaI/EcoRI sites of Flag-HA-pcDNA3.1 vector (Addgene #52535). Fragments of *Dnmt1* (2–290, 291–603, 604–1124, 1125–1610) were PCR-amplified using the full-length pLVX-Dnmt1 plasmids and sub-cloned into XbaI/HindIII sites of Flag-HA-pcDNA3.1 vector (Addgene #52535).

siRNAs, miRNA inhibitors, overexpressing plasmids, and relative controls were transfected into day 4 differentiated BAT1 brown adipocytes using Lipofectamine RNAiMax transfection reagent (ThermoFisher Scientific) or Amaxa Nucleofector II Electroporator (Lonza) with an Amaxa cell line nucleofector kit L according to the manufacturer's instructions (Lonza) as we previously described[12,13]. Briefly, at days 4 of differentiation, cells ($2 \times 10^6$ cells/sample) were trypsinized and centrifuged at $90 \times g$ for 5 min at room temperature. Cells were then resuspended in nucleofector solution (100 μl/sample) with 20 pmol of different siRNAs or plasmids and seeded into 24-well plates or 100 mm dishes. Cells were harvested 2 days after for further analysis.

For oil red O staining assays, differentiated BAT1 cells were fixed with 10% formalin and washed twice with ddH2O. Cells were then incubated with 0.05% oil red O (Sigma: 00625–25G) working solution as we described previously[11]. Samples were visualized using Nikon Eclipse E800 Microscopy.

**Measurement of oxygen consumption in brown adipocytes**. Cellular oxygen consumption in brown adipocytes was measured using a XF 96 Extracellular Flux Analyzer (Agilent, Santa Clara, CA) as we previously described[111]. The measurement started with basal respiration recording followed by the addition of a sequential reagents including oligomycin for inhibition of the coupled respiration and FCCP for maximal respiration.

**RNA-sequencing analysis**. Total RNA was isolated from iBAT as described above and was submitted to Beijing Genomics Institute (BGI, Shenzhen, China) for RNA-sequencing (RNAseq) analysis. Equal amount of RNAs from 6 animals/group were pooled and used for RNAseq analysis. Clean reads were aligned to the mouse reference genome (UCSC mm9, https://genome.ucsc.edu/cgi-bin/hgGateway?hgsid=1183225321_q2EV6rSrk1oEv6VFl9obRLdY9dMm). Differentially expressed genes between groups were defined as Log2 fold change ≥0.5 or ≤−0.5. The RNAseq data was also used to predict adipose tissue browning capacity with an online bioinformatic software https://github.com/PerocchiLab/ProFAT [23].

**Whole-genome DNA methylation analysis with reduced representation bisulfite sequencing (RRBS) to identify Myod1 promoter methylation**. Genomic DNA from BAT was isolated by phenol chloroform extraction and a commercial service for DNA methylation sequencing was provided by Beijing Genomics Institute (BGI) (Shenzhen, China). Equal amount of genomic DNAs from 6 animals/groups were pooled and used for RRBS analysis. According to the brochure provided by the company, the genomic DNA was digested with the methylation-insensitive restriction enzyme MspI and ligated to methylated sequencing adaptors. The ligated MSPI fragments were size-selected, treated with sodium bisulfite, amplified by PCR and constructed for library, which was sequenced. Clean reads were aligned to genome annotation datasets (University of California Santa Cruz (UCSC) Genome Browser on Mouse (NCBI37/mm9) Assembly https://genome.ucsc.edu/cgi-bin/hgGateway). The software tool Bioconductor (version 3.0), a software repository of R packages, was used to produce such annotations. The tool kit methylKit v0.9.6. was used to generate methylation report for each sample, and Log2 fold change threshold of 0.5 was used to identify Differentially Methylated Regions (DMRs) between two genotypes.

**Assay for transposase-accessible chromatin sequencing (ATAC-seq) analysis**. ATAC-seq was conducted according to the Omni-ATAC-seq protocol as described[112]. Briefly, BAT tissues were dounce-homogenized, filtered, and centrifuged in iodixanol solution to obtain nuclei. The nuclei were treated with Nextera Tn5 transposase (Illumina, San Diego, CA) for the transposition reaction, followed by DNA purification and PCR amplification with NEBNext 2X MasterMix (New England BioLabs, Ipswich, MA) and Nextera Index primers (Illumina). The ATAC libraries were further size-purified and sent to Novogene (Durham, NC) for sequencing. The ATAC-seq analysis was performed on the Galaxy server as described[112].

**Chromatin immunoprecipitation (ChIP) and Re-ChIP assays**. ChIP assays were conducted using a ChIP assay kit (Upstate) as we previously described[12,13]. Tissues were fixed and used for nuclei isolation. The nuclei were resuspended in lysis buffer and sonicated to shear DNA, followed by immunoprecipitation, elution, and analyzed by real-time PCR using SYBR green. Primer sequences used in this study were: *Myod1*, forward: 5′-ACTCCTATTGGCTTGAGGCG-3′, reverse: 5′-CAAG CCGTGAGAGTCGTCTT-3′; *Prdm16*, forward: 5′-ACGAAGAGGATGATGAAC ACATT-3′, reverse: 5′-TCATCTCCCTAGCATTGTCAGTT-3′.

ChIP and Re-ChIP assay was conducted as previously described[113]. Briefly, tissue fixation, nuclei isolation and DNA shearing were performed as we described[12,13]. The sheared DNA was subjected to two sequential steps of immunoprecipitation, first against PRDM16 (Sigma, SAB3500989), and second against DNMT1 (sc-271729). The enrichment of PRDM16 and DNMT1 on *Myod1* promoter was analyzed by real-time PCR using SYBR green with primers corresponding to *Myod1* promoter as described above.

**Bisulfite conversion and pyrosequencing**. The detection of DNA methylation levels at *Myod1* promoter and 5′-region was conducted as we previously described[108]. Genomic DNA was isolated by phenol/chloroform extraction, followed by bisulfite conversion with an EpiTech Bisulfite Kit (Qiagen). The primers for PCR-amplification of *Myod1* proximal promoter/5′-region and for sequencing are shown in Supplementary Table 4. Bisulfite-converted DNA (2 μg) was amplified by PCR, which was sent to EpiGenDx (Hopkinton, MA) for sequencing analysis.

**Targeted demethylation at the Myod1 promoter**. A endonuclease dead version of Cas9 (dCas9) has been engineered to be fused with the catalytic domain of the enzyme involved in DNA demethylation, TET1 (dCas9-TET1CD)[52,55]. The mammalian lentiviral vectors FUW carrying dCas9-TET1CD were purchased from

Addgene (Addgene No. 84475). Guide sequences targeting the CpG sites at the *Myod1* promoter was designed with GT-Scan website (http://gt-scan.braembl.org.au/gt-scan) and targeting or non-targeting oligos were annealed and inserted into the AarI sites of the pgRNA lentiviral vector that co-express mCherry (Addgene No. 44248). The gRNA sequence for *Myod1*-targeting oligo was: forward, 5′-caccGTAC TGTTGGGGTTCCGGAGTGG-3′, reverse, 5′-aaacCCACTCCGGAACCCCAACA GTAC-3′. The sequence for non-targeting gRNA was: forward, 5′-ttggCCCCCGG GGGAAAAATTTTT; reverse, 5′- aaacAAAAATTTTTCCCCCGGGGG-3′[55]. Lentiviruses expressing dCas9-TET1CD or gRNA-mCherry ($1 \times 10^9$ IFU/ml) was produced by Vigene Biosciences, Inc., and was bilaterally injected into iBAT according to previously published methods[55,109,114]. Briefly, a small skin incision was made above iBAT, and a series of microinjections with designated lentivirus ($1 \times 10^9$ IFU/ml) were given across five evenly distributed loci (2 μl/locus) for each iBAT pad (10 total injections per animal) and the needle left in place for 1 min following each injection to prevent efflux. After the final injection, the skin was closed with sterile wound clip staples and animals were returned to their cages. Animals were kept isolated in their home cage for 7 days for recovery.

**Statistical analysis**. Data were expressed as mean ± SEM. All graphs were made with GraphPad Prism (v9.1.2). Data normality and homogeneity were tested using Kolmogorov-Smirnov (K-S) and Levene's tests, respectively. For parametric data, statistical significance among groups was analyzed by unpaired Student's *t*-test, one-way analysis of variance (ANOVA) followed by Fisher's Least Significant Difference (LSD) post-hoc analysis or one-way ANOVA with repeated measures; for non-parametric data, statistical significance among groups was analyzed by Mann–Whitney's *U* test or Kruskal–Wallis non-parametric ANOVA by rank; and correlation between two groups was analyzed by Spearman's Rank Correlation Coefficient test. Detailed statistical analysis methods were indicated in figure legends under each figure. Statistical significance was accepted at $p < 0.05$, and a tendency of statistical significance was accepted at $p < 0.1$. Student's *t*-test was performed using Microsoft Excel software from Microsoft Office Professional Plus 2019 (version 1808, build 10377.20023, Microsoft Corporation, Redmond, WA, USA); all other statistical tests were performed using SPSS software (version 16.0, SPSS Inc, Chicago, IL, USA).

**Reporting summary**. Further information on research design is available in the Nature Research Reporting Summary linked to this article.

## Data availability

The RNAseq, RRBS, and ATAC-seq data have been deposited to Gene Expression Omnibus (GEO) database with the accession code GSE175608. All other data generated or analyzed during this study are included in this published article (and its supplementary information files). Source data files are included with the paper. The uncropped gel images are included in the source data file. Source data are provided with this paper.

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

## Acknowledgements

This work is supported by NIH grants R01DK107544, R01DK118106, and R01DK125081, and American Diabetes Association (ADA) grant 1-18-IBS-260 to B.X.; NIH grants R01DK115740 and R01DK118106, and ADA grant 1-18-19 IBS-348 to H.S.; NIH grant R01DK116496 and ADA grant 1-18-IBS-346 to L.Y.

## Author contributions

F.L. performed most of the experiments and data analysis; J.J. assisted F.L. in these experiments; M.M. performed metabolic cage studies in wild-type C57BL/6J mice on chow or HFD; X.C., Q.C., R.W., and assisted in various experiments; Z.C., Y.P., and H.D.S. performed bioinformatics analysis of RNAseq and RRBS data; L.Y. contributed to study design, technical inputs and review/edits on the manuscript; B.X. and H.S. conceived and designed study and wrote the manuscript.

## Competing interests

The authors declare no competing interests.

## Additional information

**Peer review information** *Nature Communications* thanks Jiandie Lin and the other anonymous reviewer(s) for their contribution to the peer review this work. Peer reviewer reports are available.

