## [Peer Review File · Nature Communications]

Reviewers' Comments:

Reviewer #1:

Remarks to the Author:

In this manuscript, Shi H and colleagues describes an intriguing phenomenon of myogenic gene activation in brown fat during diet-induced obesity. The authors identified the histone demethylase UTX and the DNA methyltransferase DNMT1 as two important regulators of this brown adipocyte-myocyte gene dysregulation. Conditional inactivation of UTX or DNMT1 specifically in brown adipocytes resulted in obesity and impaired systemic metabolic homeostasis associated with activation of myogenic gene program and repression of the thermogenic function. The authors further demonstrated that MyoD1 as a likely mediator of these regulatory effects. The observations that brown adipocytes activate myogenic gene program during obesity are interesting. The in vivo findings and mechanistic work largely support the overall conclusion. However, several major concerns should be addressed.

1. While the effects of UTX and DNMT1 deficiency on brown fat function and systemic metabolic phenotype were impressive, it is unlikely that they are mediated by UCP1-mediated uncoupling. In fact, complete UCP1 deficiency only elicits modest effects on systemic energy balance. The authors should explore whether UCP-1 independent thermogenic pathways, particularly creatine cycling and calcium futile cycle, are altered in these mouse models. In addition, the secretion of brown fat adipokines, such as Nrg4, PM20D1, and myostatin, should be investigated and discussed.
2. Is induction of myocyte genes in brown fat and dysregulation of UTX/DNMT1 expression observed in genetic obesity ob/ob or db/db mice? What are potential pathophysiological signals that regulate this aberrant gene induction? Some studies along this line will provide additional insights into how this brown adipocyte-myocyte dysregulation may mediate the effects of obesity on thermogenesis and energy balance. It would be also helpful to discuss the implications of these findings in human obesity and metabolic disease.
3. Is there evidence for a role of UTX and DNMT1 in beige fat induction in response to chronic cold exposure or adrenergic stimulation?
4. Is overexpression of MyoD1 in brown preadipocytes sufficient to suppress the thermogenic gene activation following adipogenic induction?

Reviewer #2:

Remarks to the Author:

In the current manuscript, the authors have investigated a potential mechanism by which tetratricopeptide/ lysine (K)-specific demethylase 6A (Utx/Kdm6a), a histone demethylase, may regulate thermogenic activity of brown adipose tissue and systemic energy homeostasis through upregulation of Prdm16 that simultaneously activates thermogenic programs and suppress Myod1 expression in brown adipocytes. They show that mice deficient in brown adipocyte Utx have decreased thermogenic activity and aberrant activation of myogenic gene expression that can lead to impaired systemic energy homeostasis in both lean and diet-induced obesity. With several experimental data, they tried to conclude that brown adipocyte Dnmt1 plays a key role to mediate anti-myogenic remodeling role for Utx by facilitating suppressive DNA methylation at Myod1 promoter. The hypothesis is potentially interesting but several key points that are neglected by the authors, make it difficult to feel confident in the conclusions that they have tried to reach. First, the authors argue that Dnmt1 is involved in Prdm16-mediated myogenic gene suppression but not thermogenic gene activation. If so, it is expected that mice deficient in brown adipocyte Dnmt1 show increased myod1 expression with unchanged thermogenic activity. However, Dnmt1 depletion in brown adipocytes led to increased myod1 expression as well as decreased thermogenic gene expression, suggesting that Dnmt1 is not only involved in the control of myod1 expression but also thermogenic gene expression through alternative mechanisms rather than utx-prdm16 axis. Accordingly, in contrast to the author's hypothesis, these data indicate that thermogenic defects in dnmt1 deficient mice and utx deficient mice would be derived from different molecular mechanisms. Second, previously, the authors have shown that brown adipocyte Utx potentiates epigenetic reprogramming of PGC1a promoter by interacting with CBP and inducing open chromatin structure, which consequently leads to thermogenic activation in brown adipocytes (Zha et al., 2015). Given both PGC1a and PRDM16 are crucial regulators of

thermogenic activity and they can regulate each other's activity in brown adipocytes, these raise a question as to the relative contribution of Prdm16 to altered thermogenic activity in Utx KO brown adipocytes. Therefore, the work does not seem to constitute a sufficient conceptual advance from previous their findings. Additionally, their findings in the current manuscript are somehow contradictory to those by Ota et. al. 2019 (Ota et al., 2019), which generated adipocyte-specific Utx knockout mice and found opposite phenotype in response to diet-induced obesity. In particular, despite of Utx deletion in both white and brown adipose tissue in this adipocyte-specific Utx knockout mice, these mice did not show changes in body weight, fat mass, histological characteristics of all major fat depots (WAT and BAT) and systemic energy metabolism under normal chow condition. Therefore, this point should be addressed in detail to support drawn conclusions in the current study. Collectively, these major issues make the manuscript very descriptive and provides very limited insight into how utx actually epigenetically regulate thermogenesis and myogenic remodeling in obesity.

Other major concerns:

1. In Figure 1, the authors analyzed metabolic phenotypes of male UTXKO mice whereas female D1KO mice were subjected to analysis of metabolic phenotypes in Figure 4. Given it has been suggested that gender difference can affect metabolic phenotypes (Valencak et al., 2017), the use of different gender mouse to support their conclusion on key role of UTX-DNMT1 axis in myogenic remodeling would not be appropriate.
2. In supplemental Figure 6F, the authors showed the inverse correlation of UCP1 expression with various myogenic genes in iBAT. However, correlation analysis of UTX1 expression with these myogenic genes was not provided in the current manuscript. Because the major hypothesis of this study is the negative effect of UTX1 on myogenic gene expression, data demonstrating correlation between UTX1 and myogenic genes would be helpful as well.
3. In Figure 2, most experiments were done with the use of iBAT and the authors appear to suggest that such increase is derived from brown adipocytes in iBAT. However, given the expression of myogenic genes is relatively higher in stromal vascular fraction (SVF), especially brown adipocyte precursors, of iBAT as compared to brown adipocytes and HFD induces dynamic changes not only in brown adipocyte but also SVF of iBAT, it would not be clear whether HFD-induced increases in myogenic gene expression in iBAT would be attributable to SVF or brown adipocytes. Validation of the major cellular contributor of myogenic gene expression induced by HFD or UTX knockout would strengthen the argument. Also, it would be helpful to determine whether UTX regulated-BAT-to-myocyte remodeling via a cell autonomous manner.
4. In Figure 7F-H, the authors showed that Prdm16 knockdown reduced DNA methylation and Dnmt1 binding at Myod1 promoter. Whether such changes in DNA methylation and Dnmt1 binding would impact on Myod1 expression should be determined.
5. Co-IP western blots in Figure 7I-K are very poor quality and appropriate controls (e.g, input) are missing. The interaction between Prdm16 and Dnmt1 should be validated through well-controlled reciprocal immune-precipitation and by demonstrating nuclear colocalization of Prdm16 and Dnmt1. Additionally, if Prdm16 and Dnmt1 interacts, how Prdm16 increases Dnmt1 binding to Myod1 promoters? Does Prdm16 guide Dnmt1 to Myod1 promoter or control Dnmt1 stability so that more Dnmt1 can bind to Myod1 promoter instead of being degraded? Previous studies on Dnmt1 stability have demonstrated that NTD domain of Dnmt1, which may be responsible for interaction of Dnmt1 with Prdm16, is important for Dnmt1 stability through modulating interaction with SET7 and AKT1(Esteve et al., 2011). This study implies that Prdm16 and SET7 may compete for binding to Dnmt1, and thus increased Prdm16 level would lead to prevent SET7-mediated Dnmt1 degradation. Therefore, addition of data on Dnmt1 stability and Dnmt1 binding to Myod1 promoter (e.g, EMSA assay) in response to altered Prdm16 expression is needed to provide more mechanistic insights into physiological role of Utx-Dnmt1 interaction.
6. What is the mechanism whereby Dnmt1-mediated methylation decreases Myod1 expression? What are the effects of Dnmt1-mediated DNA methylation on bindings of related transcription factor(s) and/or histone modifications, which are engaged in the activation of Myod1 transcription in brown adipocytes?

Minor concerns:

1. There are no scale bars in most histology and image data. The authors need to include proper scale bars in desired data (e.g., Figure1D, 2F, 2J, 3C, 3J, 6H).
2. There are no information on molecular weight of target bands in all western blot data. The authors should provide appropriate information in all western blot data.
3. In line 358, there is a typo error. Myd1 needs to change to Myod1.

References

- Esteve, P.O., Chang, Y., Samaranayake, M., Upadhyay, A.K., Horton, J.R., Feehery, G.R., Cheng, X., and Pradhan, S. (2011). A methylation and phosphorylation switch between an adjacent lysine and serine determines human DNMT1 stability. *Nat Struct Mol Biol* 18, 42-48.
- Ota, K., Komuro, A., Amano, H., Kanai, A., Ge, K., Ueda, T., and Okada, H. (2019). High Fat Diet Triggers a Reduction in Body Fat Mass in Female Mice Deficient for Utx demethylase. *Sci Rep* 9, 10036.
- Valencak, T.G., Osterrieder, A., and Schulz, T.J. (2017). Sex matters: The effects of biological sex on adipose tissue biology and energy metabolism. *Redox Biol* 12, 806-813.
- Zha, L., Li, F., Wu, R., Artinian, L., Rehder, V., Yu, L., Liang, H., Xue, B., and Shi, H. (2015). The Histone Demethylase UTX Promotes Brown Adipocyte Thermogenic Program Via Coordinated Regulation of H3K27 Demethylation and Acetylation. *J Biol Chem* 290, 25151-25163.

Reviewer #1

1. While the effects of UTX and DNMT1 deficiency on brown fat function and systemic metabolic phenotype were impressive, it is unlikely that they are mediated by UCP1-mediated uncoupling. In fact, complete UCP1 deficiency only elicits modest effects on systemic energy balance. The authors should explore whether UCP-1 independent thermogenic pathways, particularly creatine cycling and calcium futile cycle, are altered in these mouse models. In addition, the secretion of brown fat adipokines, such as Nrg4, PM20D1, and myostatin, should be investigated and discussed.

Thank you for the helpful suggestions. We agree that the obesity phenotypes in our animal models may not be mediated solely by UCP1-mediated uncoupling, as complete UCP1 deficiency only elicits modest effects on energy homeostasis. Thus, we explored whether UCP-1 independent thermogenic pathways could be involved in regulating the phenotypes in our animal models. We found that the expression of SERCA2b/ATP2a2 was not altered in iBAT of UTXKO mice, and the expression of creatine kinase (*Ckm*) was even increased in iBAT of UTXKO mice under either the HFD or cold challenge along with the upregulation of myogenic marker expression (**Supplemental Figure 5**). In addition, similar gene expression patterns of creatine kinase and *Atp2a2* was also observed in iBAT of D1KO mice (**Figure 4B and Supplemental Figure 15A**). Thus, we think these UCP1-independent thermogenic pathways, including the creatine cycling and calcium futile cycle, are not involved in mediating the obesity-prone phenotypes observed in UTXKO and D1KO mice. It is possible that UTX or DNMT1 deficiency in brown fat alters the entire thermogenic program beyond UCP1, thus resulting in a profoundly obesity-prone phenotype. Alternatively, other UCP1-independent mechanisms may be involved, which requires further identification. These results were described and discussed in page 7, lines 156-163, and page 15, lines 360-365.

In addition, we have also explored whether batokines secreted from brown fat, including the recently identified neuregulin 4 (*Nrg4*), peptidase M20 domain containing 1 (*Pm20d1*) and myostatin (*Mstn*), could be involved in regulating whole body insulin sensitivity and metabolic function in other organs in our KO animal models. However, we did not observe significant alterations of the expression of these batokines in iBAT of UTXKO (**Supplemental Figure 4**) and D1KO (**Supplemental Figure 15B-C**) animals. Thus, we think these batokines may not be involved in the regulation of whole body insulin sensitivity and metabolism in our animal models. These results are now described and discussed in pages 6-7, lines 142-155, and page 15, lines 365-366.

2. Is induction of myocyte genes in brown fat and dysregulation of UTX/DNMT1 expression observed in genetic obesity *ob/ob* or *db/db* mice? What are potential pathophysiological signals that regulate this aberrant gene induction? Some studies along this line will provide additional insights into how this brown adipocyte-myocyte dysregulation may mediate the effects of obesity on thermogenesis and energy balance. It would be also helpful to discuss the implications of these findings in human obesity and metabolic disease.

Thank you for the comments. This is an important question to assess whether the BAT-to-myocyte switch observed in the UTXKO, D1KO and HFD-fed mice is a common feature of obesity. We have measured myogenic gene expression in iBAT of 3-month-old *ob/ob* mice and their control +/- littermates. We observed a similarly upregulated myogenic gene expression in iBAT of *ob/ob* mice,

accompanied by down-regulation of thermogenic gene expression (**Supplemental Figure 12**, and pages 11-12, lines 280-287).

To answer the question about the “potential pathophysiological signals that regulate this aberrant gene induction” in obesity, we think there are several possibilities.

First, we have previously found that UTX mRNA level was down-regulated in iBAT of HFD-fed mice (Zha et al, JBC 2015). Since UTX protein level could be more important than mRNA expression, we have measured UTX protein levels in iBAT of HFD-fed mice. We found that UTX protein level was significantly reduced in iBAT of mice after 12 weeks of HFD feeding (**Figure 3**). Thus, we think reduced UTX levels that occur during obesity could explain the myogenic remodeling and suppression of thermogenic program observed in iBAT during obesity development. UTX was predicted to be ubiquitinated at lysine 225¹. Thus, it is possible that increased UTX ubiquitination could occur in obesity and contribute to decreased UTX protein levels in iBAT. Further studies are required to investigate whether UTX ubiquitination occurs in iBAT of HFD-fed mice that may contribute its down-regulation. These data have been described and discussed in page 10, lines 243-249, and page 29, lines 740-743.

Alternatively, as we discussed in page 29, lines 743-758, recent data suggest that UTX serves as a cellular oxygen sensor, while hypoxia that reduces the level of oxygen for UTX to sense, cause persistent increase in H3K27 methylation in key regulators of cellular differentiation, which in turn blocks cellular differentiation². Hypoxia develops in obese white adipose tissue with the expansion of tissue mass, and contributes to adipose tissue dysfunction, including inflammation, fibrosis and insulin resistance³. Recent data also suggest that obesity induces capillary rarefaction and functional hypoxia in BAT, leading to a BAT “whitening” phenotype⁴. Thus, it is possible that hypoxic condition induced in obese BAT may inhibit UTX activity, causing a persistent increase in H3K27 methylation at promoters of key regulators of differentiation. This in turn reverses these brown adipocytes to a less-differentiated, precursor-like phenotype, leading to a loss of brown adipocyte thermogenic feature and simultaneous up-regulation of myogenic gene expression in Utx-deficient brown adipocytes, as observed in the current study.

About the comment *“It would be also helpful to discuss the implications of these findings in human obesity and metabolic disease”*, we have added the following discussion in the manuscript in pages 27-28, lines 684-693:

It has been recently demonstrated that adult humans also have functional BAT that is inversely correlated with body weight^{5, 6, 7, 8, 9, 10, 11}, and repeated cold exposure treatments activate human BAT that is associated with an increase in energy expenditure^{12, 13}. Thus, BAT may present a promising target for the treatment of obesity and other metabolic diseases in humans. Interestingly, H3K27me3, the target of UTX, is an inheritable epigenetic pattern that is associated with obesity and type 2 diabetes¹⁴. In addition, Kabuki syndrome is a rare genetic disease with obesity and type 2 diabetes as frequently reported symptoms among others, and Utx is reported as one of the genes to be mutated^{15, 16, 17, 18, 19, 20}. On the other hand, the association of Dnmt1 polymorphism with obesity has also been reported in human population²¹. Thus, it is possible that the UTX/DNMT1 pathway may also regulate obesity and associated metabolic diseases in humans.

3. Is there evidence for a role of UTX and DNMT1 in beige fat induction in response to chronic cold exposure or adrenergic stimulation?

To answer this question, we have studied gene expression and UCP1 immunostaining in inguinal WAT and epididymal WAT from D1KO mice after a chronic 7-day cold exposure. Interestingly, we found that in iWAT, whereas the expression of *Ucp1* and other thermogenic genes were either tended to increase or moderately increased, UCP1-immunostaining clearly showed an increase in beige adipocyte formation in iWAT of D1KO mice compared to fl/fl littermates after a 7-day cold exposure (**Suppl. Fig 19A-B**). In addition, thermogenic gene expression and beige adipocyte formation were also increased in eWAT of D1KO compared to that of fl/fl littermates after the chronic cold exposure (**Suppl. Fig 19C-D**).

These data indicate an intriguing dissociation of thermogenesis between traditional brown adipocytes in iBAT and beige adipocytes identified in WAT. Recent data suggest that BAT and WAT are derived from different developmental origins. Brown adipocytes from iBAT and skeletal muscle cells share a common developmental lineage and are derived from precursor cells that express Myf5, whereas most white adipocytes from eWAT and iWAT come from a different lineage origin that does not express Myf5. Thus, it is possible that the role of DNMT1 in the determination of BAT-muscle switch may be specific to the Myf5-positive lineage cells in iBAT, but not the Myf5-negative lineage cells in eWAT and iWAT. The increased beiging in iWAT and eWAT of D1KO mice in response to cold is possibly a compensatory adaptation to reduced thermogenic function in iBAT.

These results are now described and discussed in pages 17, lines 407-423.

4. Is overexpression of MyoD1 in brown preadipocytes sufficient to suppress the thermogenic gene activation following adipogenic induction?

We have overexpressed MyoD1 in brown preadipocytes and found it sufficient to suppress the thermogenic gene activation following adipogenic induction both at basal level and after the β -adrenergic stimulation by isoproterenol. These data are presented in **Fig 7L-M** and described in page 20, lines 492-497.

Reviewer #2

1. First, the authors argue that *Dnmt1* is involved in *Prdm16*-mediated myogenic gene suppression but not thermogenic gene activation. If so, it is expected that mice deficient in brown adipocyte *Dnmt1* show increased *myod1* expression with unchanged thermogenic activity. However, *Dnmt1* depletion in brown adipocytes led to increased *myod1* expression as well as decreased thermogenic gene expression, suggesting that *Dnmt1* is not only involved in the control of *myod1* expression but also thermogenic gene expression through alternative mechanisms rather than *utx-prdm16* axis. Accordingly, in contrast to the author's hypothesis, these data indicate that thermogenic defects in *dnmt1* deficient mice and *utx* deficient mice would be derived from different molecular mechanisms.

We agree with the reviewer that in the D1KO mice, we observe not only the increased expression of myogenic genes, but also a significant down-regulation of thermogenic program. The up-regulation of myogenic gene expression in *Dnmt1*-deficient brown adipocytes could be explained by

reduced *Myod1* promoter DNA methylation due to *Dnmt1* deficiency. However, how up-regulated *Myod1* expression in *Dnmt1*-deficient brown adipocytes results in reduced thermogenic gene expression is less clear. Thus, we have further explored several possibilities that could explain the down-regulation of thermogenic gene expression in *Dnmt1*-deficient brown adipocytes.

Recent data suggest that *Myod1* represses *Prdm16* in muscle progenitor cells via interaction with the E2F transcription factor 4 (E2F4)/RB transcriptional corepressor like 1 (RBL1/p107)/ RB transcriptional corepressor like 2 (RBL2/p130) transcription repressor complex at *Prdm16* promoter²². Thus, we further explored whether MYOD1 could repress *Prdm16* via this potential pathway in BAT1 cells. We found that overexpressing *Myod1* in BAT1 cell led to increased binding of E2F4, RBL1 and RBL2 at *Prdm16* promoter (**Suppl. Fig 24A-C**). However, knocking down of *E2f4*, *Rbl1* or *Rbl2* in BAT1 cells (**Suppl. Fig 24D-F**) did not significantly change basal expression of BAT-specific genes, nor did they reverse *Myod1* overexpression-induced suppression of BAT-specific gene expression when *E2f4*, *Rbl1* or *Rbl2* were knocked down individually or in combination (**Suppl. Fig 24G-J**). These data suggest that unlike its role in muscle progenitor cells, the E2F4/RBL1/RBL2 repressor complex does not play an important role in mediating MYOD1's suppression of BAT-specific gene expression in BAT1 brown adipocytes.

Alternatively, microRNA-1 (miR-1), miR-206, miR-133a and miR-133b are specifically expressed in skeletal muscle that are important in regulating muscle development^{23,24,25,26}, and their expression is regulated by MYOD1^{24,25}. Recently, it has been reported that miR-133a/133b negatively regulates BAT-specific gene expression in muscle satellite cells by specifically targeting *Prdm16*²⁶. Interestingly, we found that the expression of these microRNAs was significantly increased in iBAT of D1KO mice (**Fig 10G**) and in BAT1 cells overexpressing *Myod1* (**Fig 10H**). In addition, *Dnmt1* knocking down in BAT1 cells also significantly up-regulated *miR-133a* expression (**Fig 10I**). Importantly, knocking down *Dnmt1* in BAT1 cells significantly suppressed BAT-specific gene expression, including *Ucp1*, *Prdm16* and *Pgc1b*, which was reversed by *miR-133a* knocking down (**Fig 10I-J**). Thus, our data suggest that increased *Myod1* expression in *Dnmt1*-deficient brown adipocytes results in upregulation of miR133, which in turn suppresses *Prdm16* expression, leading to down-regulation of thermogenic gene expression.

These results are now presented in **Fig 10 G-I** and **Suppl. Fig 24**, and described and discussed in pages 25-26, lines 616-650.

2. Second, previously, the authors have shown that brown adipocyte *Utx* potentiates epigenetic reprogramming of *PGC1a* promoter by interacting with CBP and inducing open chromatin structure, which consequently leads to thermogenic activation in brown adipocytes (Zha et al., 2015). Given both *PGC1a* and *PRDM16* are crucial regulators of thermogenic activity and they can regulate each other's activity in brown adipocytes, these raise a question as to the relative contribution of *Prdm16* to altered thermogenic activity in *Utx* KO brown adipocytes. Therefore, the work does not seem to constitute a sufficient conceptual advance from previous their findings.

We agree with the reviewer's comments. Our previous data suggested that UTX promotes brown adipocyte thermogenic program via coordinated regulation of H3K27 demethylation and acetylation at *Pgc1a* promoter²⁷; whereas PRDM16 is critical in controlling BAT-muscle switch²⁸. To further determine the molecular mechanism underlying the BAT-to-myocyte remodeling phenotype in BAT of UTXKO and D1KO mice, we performed assay for transposase-accessible chromatin sequencing (ATAC-seq) analysis in iBAT of UTXKO and fl/fl littermates fed HFD for 12 weeks. Our ATAC-seq data

demonstrated that deleting *Utx* in brown fat resulted in a more closed chromatin structure at both *Prdm16* (Fig 9B) and *Pgc1α* (Suppl. Fig 23A) loci, which could contribute to the suppressed expression of these genes in iBAT of UTXKO mice (Fig 1F). Our data indicate that both *Prdm16* and *Pgc1α* could be downstream targets of UTX via modulating their chromatin structure. Since PRDM16 regulates BAT-muscle switch in brown adipose tissue²⁸, we thus further studied whether PRDM16 could mediate the BAT-to-myocyte remodeling observed in HFD-fed, UTXKO and D1KO mice. These data have been described in page 21-22, lines 533-553.

Although PRDM16 have been studied extensively in regulating BAT-myocyte fate determination, the mechanism involved is not fully understood. Thus, our data will advance the field by discovering an epigenetic signaling network involving UTX-PRDM16-DNMT1-MYOD1 that mediates PRDM16's effect on BAT-myocyte fate determination in brown adipocytes.

3. Additionally, their findings in the current manuscript are somehow contradictory to those by Ota et al. 2019 (Ota et al., 2019), which generated adipocyte-specific Utx knockout mice and found opposite phenotype in response to diet-induced obesity. In particular, despite of Utx deletion in both white and brown adipose tissue in this adipocyte-specific Utx knockout mice, these mice did not show changes in body weight, fat mass, histological characteristics of all major fat depots (WAT and BAT) and systemic energy metabolism under normal chow condition. Therefore, this point should be addressed in detail to support drawn conclusions in the current study. Collectively, these major issues make the manuscript very descriptive and provides very limited insight into how utx actually epigenetically regulate thermogenesis and myogenic remodeling in obesity.

Thanks for this comment. Along the course of our study, a recent report brought us an attention that deletion of *Utx* in all adipocytes including both brown and white adipocytes prevents HFD-induced obesity²⁹. The exact reason for the discrepancy between this paper's observation and ours is not clear, but may lie within the different models being tested and the realm of adipocytes (for example, brown or white adipocytes) that had *Utx* deletion. For one, Ota et al generated the genetic model with *Utx* deficiency in all adipocytes by crossing *Utx* floxed mice with aP2-Cre mice. It is noteworthy that employing aP2-Cre line may also result in deletion of *Utx* in macrophages due to the presence of aP2 in macrophages³⁰. It is possible that *Utx* deficiency in macrophages may alter inflammatory status that in turn affects adipocyte metabolism through a paracrine action due to the infiltration of macrophage into adipose tissue during the development of obesity³¹. Indeed, specific deletion of *Utx* in macrophages by lysozyme-Cre has been shown to increase M1 macrophage formation, inhibit white adipocyte differentiation and promote brown adipocyte thermogenesis, leading to decreased adiposity in mice³². Thus, the obesity-resistant phenotype observed by Ota et al²⁹ may be due to the aP2-Cre used that resulted in *Utx* deletion also in macrophages. In addition, we also found that whereas mice with mature brown adipocyte-specific deletion of *Utx* using the *Ucp1*-Cre exhibited a BAT-to-myocyte remodeling and were prone to diet-induced obesity, mice with *Utx* deletion in *Myf5*+ cells did not show any phenotypic differences when fed HFD, despite the fact that brown adipocytes derive from *Myf5*+ lineage precursors²⁸. Thus, the differences in the Cre line used may contribute to the differential phenotypes observed between our animal models and those reported in Ota et al²⁹. This discussion was added in pages 30-31, lines 759-778.

Hopefully these additional data and discussions (including the ones added below) could convince the reviewers that our manuscript provides a mechanistic point about how UTX regulates thermogenesis and myogenic remodeling in obesity.

Other major concerns:

1. In Figure 1, the authors analyzed metabolic phenotypes of male UTXKO mice whereas female D1KO mice were subjected to analysis of metabolic phenotypes in Figure 4. Given it has been suggested that gender difference can affect metabolic phenotypes (Valencak et al., 2017), the use of different gender mouse to support their conclusion on key role of UTX-DNMT1 axis in myogenic remodeling would not be appropriate.

Thank you for pointing this out. Although we presented data on female D1KO mice in Figure 4 (**now Figure 5**), both male and female D1KO mice exhibited similar metabolic phenotypes including diet-induced obesity and insulin resistance. We presented data on male D1KO mice that showed a slightly milder but still significant obesity-prone phenotype in **Supplemental Figure 16 and 18**. In addition, we found that both male and female UTXKO mice showed an obesity-prone phenotype when fed HFD, albeit with much milder phenotype observed in female UTXKO mice. The data on female UTXKO mice is now added in **Supplemental Figure 6 and page 7, lines 164-171**. Thus, our data suggest sexual dimorphism is more evident in UTXKO mice than in D1KO mice. The exact reason is not clear.

Sexual dimorphism frequently occurs in metabolic phenotypes of both humans and rodents. For one, males and females have different fat composition and distribution in humans^{33,34}. This might be due to differential lipid metabolism between the two genders³⁵. Another potential mechanism may be attributed to the sex hormone estrogen and its receptors that have been shown to play a pivotal role in various metabolic pathways³⁶.

Our data in the current study suggest that UTX may potentially be acting as an upstream regulator of DNMT1. Thus, UTX signaling may be more susceptible to the modification by other genetic or environmental cues than the downstream DNMT1. It is possible these factors regulate UTX downstream signaling other than DNMT1 in UTXKO mice that may contribute to the phenotype difference observed in UTXKO and D1KO mice. These discussions have been added in page 31, lines 779-790.

2. In supplemental Figure 6F, the authors showed the inverse correlation of UCP1 expression with various myogenic genes in iBAT. However, correlation analysis of UTX1 expression with these myogenic genes was not provided in the current manuscript. Because the major hypothesis of this study is the negative effect of UTX1 on myogenic gene expression, data demonstrating correlation between UTX1 and myogenic genes would be helpful as well.

Thank you for pointing this out. Previously, we found that UTX mRNA level was down-regulated in iBAT of HFD-fed mice²⁷. Since UTX protein level could be more important than mRNA expression, we have now measured UTX protein levels in iBAT of HFD-fed mice and found that UTX protein level was significantly decrease in iBAT of mice fed HFD for 12 and 24 weeks (**Fig 3A**). Further analysis also revealed a reciprocal pattern of UTX protein levels and myogenic marker gene expression in iBAT of

HFD-fed mice, with a gradual decline of UTX protein level that corresponded to a reciprocal increase of myogenic marker gene expression following HFD feeding (**Fig 3B**). An inverse correlation also existed between UTX protein levels and myogenic marker gene expression in iBAT when analyzed from both chow- and HFD-fed mice (**Fig 3C**). These data are now presented in **Fig. 3A-C**, and described in page 10, lines 243-249.

3. In Figure 2, most experiments were done with the use of iBAT and the authors appear to suggest that such increase is derived from brown adipocytes in iBAT. However, given the expression of myogenic genes is relatively higher in stromal vascular fraction (SVF), especially brown adipocyte precursors, of iBAT as compared to brown adipocytes and HFD induces dynamic changes not only in brown adipocyte but also SVF of iBAT, it would not be clear whether HFD-induced increases in myogenic gene expression in iBAT would be attributable to SVF or brown adipocytes. Validation of the major cellular contributor of myogenic gene expression induced by HFD or UTX knockout would strengthen the argument. Also, it would be helpful to determine whether UTX regulated-BAT-to-myocyte remodeling via a cell autonomous manner.

We agree with reviewer's comments. Recent data suggest that both stromal vascular fraction (SVF) cells and brown adipocytes contribute to Myf5+ cell populations in lineage tracing studies³⁷. In addition, it has been reported that brown preadipocytes express a prominent myogenic transcriptional signature, which is diminished upon brown adipocyte differentiation³⁸. Thus, to investigate whether the observed BAT-to-myocyte remodeling is derived from brown adipocyte or cells from SVF, we isolated primary brown adipocytes and SVF cells from iBAT of C57BL/6J mice fed chow or HFD for 24 weeks. Similar to previous report³⁸, we found that myogenic markers were expressed in both SVF cells and primary brown adipocytes; the expression of myogenic genes was relatively higher in SVF cells than that of primary brown adipocytes isolated from mice fed chow diet (Suppl. Fig 10A). However, while HFD either did not change or only induced a mild increase in myogenic marker expression in SVF cells, HFD seemed to induced more profound myogenic marker expression in primary brown adipocytes than in SVF cells to a level similar or higher than that of SVF levels (Suppl. Fig 10A). The increase in myogenic gene expression in primary brown adipocytes was more evident when gene expression was normalized to that of chow values in each of the SVF and adipocyte groups (Suppl. Fig 10B). Thus, our data suggest a prominent BAT-to-myocyte remodeling in primary brown adipocytes upon HFD feeding, which may primarily contribute to the BAT-to-myocyte switch observed in iBAT from HFD-fed mice. These data are now presented in **Suppl. Fig 10** and described in page 11, lines 259-275.

In addition, to investigate whether UTX regulates BAT-to-myocyte remodeling via a cell autonomous manner, we knocked down *Utx* in a brown adipocyte cell line BAT1 cells^{39,40}. As expected, knocking down *Utx* in BAT1 cells significantly upregulated MyHC immunostaining (**Fig 2I**), similar to that of iBAT from UTXKO mice (**Fig 2G**), indicating UTX's regulation of BAT-to-myocyte remodeling is mediated via a cell autonomous manner. This has been presented in **Fig 2I**, and described in page 9, lines 212-216.

4. In Figure 7F-H, the authors showed that *Prdm16* knockdown reduced DNA methylation and *Dnmt1* binding at *Myod1* promoter. Whether such changes in DNA methylation and *Dnmt1* binding would impact on *Myod1* expression should be determined.

Thank you for pointing this out. We have now measured *Myod1* expression in BAT1 cells with *Prdm16* knockdown. We found that *Prdm16* knockdown in BAT1 cells reduced DNA methylation at *Myod1* promoter (now **Fig 9I**), with concomitant up-regulation of *Myod1* expression (**Fig 9J**). In addition, we found *Utx* knockdown in BAT1 cells reduced DNA methylation at *Myod1* promoter (**Fig 9G**), which was also associated with upregulation of *Myod1* expression (**Fig 9H**). These data are now presented in **Fig 9J** and **9H**, and described in page 23, lines 565-571.

5. Co-IP western blots in Figure 7I-K are very poor quality and appropriate controls (e.g, input) are missing. The interaction between Prdm16 and Dnmt1 should be validated through well-controlled reciprocal immune-precipitation and by demonstrating nuclear colocalization of Prdm16 and Dnmt1. Additionally, if Prdm16 and Dnmt1 interacts, how Prdm16 increases Dnmt1 binding to Myod1 promoters? Does Prdm16 guide Dnmt1 to Myod1 promoter or control Dnmt1 stability so that more Dnmt1 can bind to Myod1 promoter instead of being degraded? Previous studies on Dnmt1 stability have demonstrated that NTD domain of Dnmt1, which may be responsible for interaction of Dnmt1 with Prdm16, is important for Dnmt1 stability through modulating interaction with SET7 and AKT1(Esteve et al., 2011). This study implies that Prdm16 and SET7 may compete for binding to Dnmt1, and thus increased Prdm16 level would lead to prevent SET7-mediated Dnmt1 degradation. Therefore, addition of data on Dnmt1 stability and Dnmt1 binding to Myod1 promoter (e.g, EMSA assay) in response to altered Prdm16 expression is needed to provide more mechanistic insights into physiological role of Utx-Dnmt1 interaction.

We agree with reviewer's comments. We have now added input controls for the experiments studying the interactions between PRDM16 and DNMT1 (now **Fig 10B, C and D**).

We also agree that recent data suggest that the NTD domain on DNMT1 regulates its protein stability by interacting with the protein lysine methyltransferase SET7 and the serine/threonine kinase AKT1⁴¹. Thus, to study whether the observed interaction between PRDM16 and the NTD domain on DNMT1 regulates DNMT1 protein stability, we co-transfected PRDM16 and DNMT1 into HEK293T cells and treated cells with the protein translation inhibitor cycloheximide. As shown in **Fig 10E**, co-transfection of PRDM16 and DNMT1 significantly prevented DNMT1 protein degradation following cycloheximide treatment, indicating that the presence of PRDM16 increases DNMT1 protein stability, possibly through binding to the NTD domain on DNMT1.

In addition, ChIP and Re-ChIP assay on *Myod1* promoter with serial immunoprecipitation against PRDM16 and DNMT1 suggest that the presence of PRDM16 significantly increased both PRDM16 and DNMT1 binding on *Myod1* promoter (**Fig 10F**), which could in turn regulate *Myod1* promoter DNA methylation and *Myod1* expression. Thus, our data suggest that the increased binding of PRDM16 with DNMT1 at *Myod1* promoter when PRDM16 is present is probably due to two mechanisms, first, PRDM16 could increase DNMT1 protein stability via binding to the NTS domain on DNMT1, as shown in **Fig 10E**; and second, PRDM16 could increase the recruitment of DNMT1 to the *Myod1* promoter, as shown in the ChIP and Re-ChIP assay in **Fig 10F**. These data are described in pages 24-25, lines 600-615.

6. What is the mechanism whereby Dnmt1-mediated methylation decreases Myod1 expression? What are the effects of Dnmt1-mediated DNA methylation on bindings of related transcription factor(s) and/or histone modifications, which are engaged in the activation of Myod1 transcription in brown adipocytes?

Thanks for this comment. Changes in DNA methylation have been shown to modulate histone methylation, which may act cooperatively to influence chromatin structure and thereby regulate gene expression⁴². We therefore interrogated the effects of DNMT1-mediated DNA methylation on histone modifications at the *Myod1* promoter by surveying the status of several common histone marks using ChIP assays. We found that Dnmt1 knockdown markedly reduced histone repressive mark histone 3 lysine 9 trimethylation (H3K9me3) while simultaneously enhancing transcriptional active mark H3K4 acetylation (H3K4ac) (**Suppl. Fig 25A-B**), without affecting other histone marks such as H3K9ac, H3K4me3 and H3K27me3 (**Suppl. Fig 25C-E**). The reciprocal changes of decreased H3K9me3 and increased H3K4ac could contribute to increased chromatin accessibility at the *Myod1* gene locus in iBAT of the UTXKO mice as revealed by ATAC-seq analysis (**Suppl. Fig 25F**), which may account for the up-regulation of *Myod1* expression that mediates BAT-to-myocyte remodeling in *Utx*- and *Dnmt1*-deficient BAT. This discussion was added in page 26, lines 651-662.

Minor concerns:

1. There are no scale bars in most histology and image data. The authors need to include proper scale bars in desired data (e.g., Figure1D, 2F, 2J, 3C, 3J, 6H).

The scale bars have been added to all histology and image data.

2. There are no information on molecular weight of target bands in all western blot data. The authors should provide appropriate information in all western blot data.

The molecular weight of the target protein bands in all western blots has been added.

3. In line 358, there is a typo error. *Myd1* needs to change to *Myod1*.

The typo has been corrected.

References:

1. Wagner SA, et al. A proteome-wide, quantitative survey of in vivo ubiquitylation sites reveals widespread regulatory roles. *Molecular & cellular proteomics* : MCP **10**, M111 013284 (2011).
2. Chakraborty AA, et al. Histone demethylase KDM6A directly senses oxygen to control chromatin and cell fate. *Science* **363**, 1217-1222 (2019).
3. Trayhurn P. Hypoxia and adipose tissue function and dysfunction in obesity. *Physiological reviews* **93**, 1-21 (2013).

4. Shimizu I, *et al.* Vascular rarefaction mediates whitening of brown fat in obesity. *J Clin Invest* **124**, 2099-2112 (2014).
5. Cypess AM, *et al.* Identification and importance of brown adipose tissue in adult humans. *N Engl J Med* **360**, 1509-1517 (2009).
6. Nedergaard J, Bengtsson T, Cannon B. Unexpected evidence for active brown adipose tissue in adult humans. *Am J Physiol Endocrinol Metab* **293**, E444-452 (2007).
7. Orava J, *et al.* Different metabolic responses of human brown adipose tissue to activation by cold and insulin. *Cell Metab* **14**, 272-279 (2011).
8. Ouellet V, *et al.* Brown adipose tissue oxidative metabolism contributes to energy expenditure during acute cold exposure in humans. *J Clin Invest* **122**, 545-552 (2012).
9. Pfannenberg C, *et al.* Impact of age on the relationships of brown adipose tissue with sex and adiposity in humans. *Diabetes* **59**, 1789-1793 (2010).
10. van Marken Lichtenbelt WD, *et al.* Cold-activated brown adipose tissue in healthy men. *N Engl J Med* **360**, 1500-1508 (2009).
11. Virtanen KA, *et al.* Functional brown adipose tissue in healthy adults. *N Engl J Med* **360**, 1518-1525 (2009).
12. van der Lans AA, *et al.* Cold acclimation recruits human brown fat and increases nonshivering thermogenesis. *J Clin Invest* **123**, 3395-3403 (2013).
13. Yoneshiro T, *et al.* Recruited brown adipose tissue as an antiobesity agent in humans. *J Clin Invest* **123**, 3404-3408 (2013).
14. Varemo L, *et al.* Type 2 diabetes and obesity induce similar transcriptional reprogramming in human myocytes. *Genome Med* **9**, 47 (2017).
15. Bereket A, Turan S, Alper G, Comu S, Alpay H, Akalin F. Two patients with Kabuki syndrome presenting with endocrine problems. *J Pediatr Endocrinol Metab* **14**, 215-220 (2001).
16. Devriendt K, Lemli L, Craen M, de Zegher F. Growth hormone deficiency and premature thelarche in a female infant with kabuki makeup syndrome. *Horm Res* **43**, 303-306 (1995).

17. Moon JE, Lee SJ, Ko CW. A de novo KMT2D mutation in a girl with Kabuki syndrome associated with endocrine symptoms: a case report. *BMC Med Genet* **19**, 102 (2018).
18. Shalev SA, *et al.* Long-term follow-up of three individuals with Kabuki syndrome. *Am J Med Genet A* **125A**, 191-200 (2004).
19. Tawa R, Kaino Y, Ito T, Goto Y, Kida K, Matsuda H. A case of Kabuki make-up syndrome with central diabetes insipidus and growth hormone neurosecretory dysfunction. *Acta Paediatr Jpn* **36**, 412-415 (1994).
20. Xin C, *et al.* Identification of novel KMT2D mutations in two Chinese children with Kabuki syndrome: a case report and systematic literature review. *BMC Med Genet* **19**, 31 (2018).
21. Chen HL, *et al.* Polymorphism of the DNA methyltransferase 1 gene is associated with the susceptibility to essential hypertension in male. *Clin Exp Hypertens* **40**, 695-701 (2018).
22. An Y, *et al.* A Molecular Switch Regulating Cell Fate Choice between Muscle Progenitor Cells and Brown Adipocytes. *Dev Cell* **41**, 382-391 e385 (2017).
23. Chen JF, *et al.* The role of microRNA-1 and microRNA-133 in skeletal muscle proliferation and differentiation. *Nat Genet* **38**, 228-233 (2006).
24. Koutsoulidou A, Mastroyiannopoulos NP, Furling D, Uney JB, Phylactou LA. Expression of miR-1, miR-133a, miR-133b and miR-206 increases during development of human skeletal muscle. *BMC Dev Biol* **11**, 34 (2011).
25. Williams AH, Liu N, van Rooij E, Olson EN. MicroRNA control of muscle development and disease. *Curr Opin Cell Biol* **21**, 461-469 (2009).
26. Yin H, *et al.* MicroRNA-133 controls brown adipose determination in skeletal muscle satellite cells by targeting Prdm16. *Cell Metab* **17**, 210-224 (2013).
27. Zha L, *et al.* The Histone Demethylase UTX Promotes Brown Adipocyte Thermogenic Program Via Coordinated Regulation of H3K27 Demethylation and Acetylation. *J Biol Chem* **290**, 25151-25163 (2015).
28. Seale P, *et al.* PRDM16 controls a brown fat/skeletal muscle switch. *Nature* **454**, 961-967 (2008).
29. Ota K, *et al.* High Fat Diet Triggers a Reduction in Body Fat Mass in Female Mice Deficient for Utx demethylase. *Scientific reports* **9**, 10036 (2019).

30. Makowski L, *et al.* Lack of macrophage fatty-acid-binding protein aP2 protects mice deficient in apolipoprotein E against atherosclerosis. *Nat Med* **7**, 699-705 (2001).
31. Lee YS, Wollam J, Olefsky JM. An Integrated View of Immunometabolism. *Cell* **172**, 22-40 (2018).
32. Chen J, *et al.* Kdm6a suppresses the alternative activation of macrophages and impairs energy expenditure in obesity. *Cell Death Differ* **28**, 1688-1704 (2021).
33. Gallagher D, Heymsfield SB, Heo M, Jebb SA, Murgatroyd PR, Sakamoto Y. Healthy percentage body fat ranges: an approach for developing guidelines based on body mass index. *Am J Clin Nutr* **72**, 694-701 (2000).
34. Palmer BF, Clegg DJ. The sexual dimorphism of obesity. *Molecular and cellular endocrinology* **402**, 113-119 (2015).
35. Schmidt SL, Bessesen DH, Stotz S, Peelor FF, 3rd, Miller BF, Horton TJ. Adrenergic control of lipolysis in women compared with men. *J Appl Physiol (1985)* **117**, 1008-1019 (2014).
36. Monteiro R, Teixeira D, Calhau C. Estrogen signaling in metabolic inflammation. *Mediators Inflamm* **2014**, 615917 (2014).
37. Shan T, Liang X, Bi P, Zhang P, Liu W, Kuang S. Distinct populations of adipogenic and myogenic Myf5-lineage progenitors in white adipose tissues. *J Lipid Res* **54**, 2214-2224 (2013).
38. Timmons JA, *et al.* Myogenic gene expression signature establishes that brown and white adipocytes originate from distinct cell lineages. *Proc Natl Acad Sci U S A* **104**, 4401-4406 (2007).
39. Rajakumari S, *et al.* EBF2 determines and maintains brown adipocyte identity. *Cell Metab* **17**, 562-574 (2013).
40. Seale P, *et al.* Transcriptional control of brown fat determination by PRDM16. *Cell Metab* **6**, 38-54 (2007).
41. Esteve PO, *et al.* A methylation and phosphorylation switch between an adjacent lysine and serine determines human DNMT1 stability. *Nat Struct Mol Biol* **18**, 42-48 (2011).
42. Cedar H, Bergman Y. Linking DNA methylation and histone modification: patterns and paradigms. *Nat Rev Genet* **10**, 295-304 (2009).

Reviewers' Comments:

Reviewer #1:

Remarks to the Author:

The authors have addressed my comments.

Reviewer #2:

Remarks to the Author:

Most concerns are properly addressed with new data in the revised manuscript. This reviewer has no further comment.

Re: Nature Communications manuscript NCOMMS-20-34403A

REVIEWERS' COMMENTS

1. Reviewer #1 (Remarks to the Author):

The authors have addressed my comments.

Answer: Thanks!

2. Reviewer #2 (Remarks to the Author):

Most concerns are properly addressed with new data in the revised manuscript. This reviewer has no further comment.

Answer: Thanks!